

# More axions from strings

**Marco Gorghetto[1], Edward Hardy[2] and Giovanni Villadoro[3]**

**1** Department of Particle Physics and Astrophysics, Weizmann Institute of Science,
Herzl St 234, Rehovot 761001, Israel
**2** Department of Mathematical Sciences, University of Liverpool,
Liverpool, L69 7ZL, United Kingdom
**3** Abdus Salam International Centre for Theoretical Physics,
Strada Costiera 11, 34151, Trieste, Italy

## Abstract

We study the contribution to the QCD axion dark matter abundance that is produced by string defects during the so-called scaling regime. Clear evidence of scaling violations is found, the most conservative extrapolation of which strongly suggests a large number of axions from strings. In this regime, nonlinearities at around the QCD scale are shown to play an important role in determining the final abundance. The overall result is a lower bound on the QCD axion mass in the post-inflationary scenario that is substantially stronger than the naive one from misalignment.

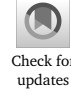

# 1  Introduction

Besides solving the strong CP problem [1] the QCD axion [2,3] may also explain the observed cold dark matter of the Universe [4–6]. In fact, if the QCD axion exists, its presence as a cold relic is almost guaranteed, unless other degrees of freedom beyond the Standard Model, if present, significantly altered the evolution of the Universe (and the physics of the axion) after reheating.

The computation of the axion relic abundance mainly depends on the relative size of the Peccei-Quinn (PQ) breaking scale $v$ compared to the largest out of the Hubble scale during inflation $H_I$ and the maximum temperature during reheating $T_{\max}$. In the so-called pre-inflationary scenario, in which $v \gtrsim \max(H_I, T_{\max})$, the PQ symmetry is broken before inflation and never restored afterwards. In this case, the relic abundance today will be different in different patches of the Universe far outside each other's cosmic horizons, so that the axion abundance in our observable Universe cannot be predicted in terms of the fundamental parameters of the theory. In this scenario, most of the experimentally allowed values of the axion mass are compatible with the observed dark matter abundance. On the other hand, in the post-inflationary scenario, in which $v \lesssim \max(H_I, T_{\max})$, the cosmological evolution of the axion field is mostly determined by the value of the axion mass, with only a mild dependence on the other model-dependent parameters. In particular, it will be the same everywhere in the Universe. In this case the totality of the dark matter can be explained by an axion only for a particular value of its mass, which is in principle calculable. In practice, computing this value is challenging, and despite various attempts over the years its determination is still afflicted by large uncertainties [7]. The main difficulties are associated to the production of axions by topological defects (global strings and domain walls) whose dynamics are nonlinear and

involve vastly different scales. Typically the thickness of domain walls and strings differ by roughly thirty orders of magnitude. This makes any attempt to directly compute the nonlinear evolution of the string-domain wall system with the physical parameters hopeless, and likewise the axion abundance that follows from the decay of these defects.

On the other hand, a lower bound on the number of axions produced could, in principle, be inferred by looking at the stage of the system's evolution that is best understood and most under control. Before the axion gets its mass at around the QCD crossover only string defects are present. Their dynamics are governed by the so-called scaling solution — an attractor of the evolution on which the properties of the string network are supposed to have simple scaling laws in terms of the relevant scales of the system. This phenomenon can be understood as an instance of self-organized criticality [8]: the expansion of the Universe keeps increasing the number of strings per Hubble patch until the string density crosses the critical point when the configuration becomes unstable. At this point strings can interact efficiently, recombining and decaying, effectively decreasing their number. The system is therefore kept at the critical point, the attractor solution, by these two competing effects. Typically the dynamics of systems at critical points simplify, becoming (approximately) scale invariant. Indeed simple scaling models have been observed to capture the main behavior of the string network [9–13], at least for local U(1) defects. For axionic strings, however, the underlying parameters that determine the dynamics are time dependent and this could cause the position and the properties of the critical point to shift. Hence the attractor solution is not expected to have exact scale invariant properties and, as we will discuss in the main text, scaling violations are indeed manifest.[1]

During the scaling regime axions are radiated from the strings, and if the properties of the network throughout this time are understood with sufficient accuracy the axions produced during this phase could also be reconstructed reliably. We should note that a huge extrapolation is still required to connect the ratio of scales that can be computed directly (slightly more than three orders of magnitude) to the physical ratio previously mentioned. However, the presence of the attractor, the fact that the scaling violations are only logarithmic and, as we will show, the fact that the final abundance is mostly determined by the qualitative features of the network, will allow us to perform such an extrapolation with some confidence.

The main inputs required for this programme are the total energy radiated from strings into axions during the scaling regime and the shape of the instantaneous axion spectrum emitted. Using energy conservation and the presence of the scaling law, the first quantity can be linked to one of the main parameters of the scaling solution: the average number of strings per Hubble patch $\xi$, which is, as we will discuss, a slowly varying function of time. Meanwhile, the spectrum is contained between an infrared (IR) cutoff set by the Hubble scale and an ultraviolet (UV) one set by the string thickness. The absence of any other scales in the problem suggests that, between these two cutoffs, the spectrum should be described by a single power law. The associated spectral index $q$ determines whether the spectrum is IR or UV dominated, i.e. whether the energy of the radiation is distributed over a large number of soft axions (for $q > 1$) or a small number of hard ones (for $q < 1$).

Although the spectrum is mostly UV dominated in the range of parameters that can be reached by present simulations [7], we find clear evidence of a non-trivial running of the spectral index, which is more compatible with an IR dominated spectrum once extrapolated to the physical parameters.

These results imply that by the time the axion mass turns on the amplitude of background axion radiation produced by strings at previous times is large. In fact the occupation number of

---

[1]Strictly speaking a non-trivial time evolution of the attractor parameters does not necessarily imply a scaling violation, but could simply indicate the presence of non-trivial critical exponents for the critical point. We however keep the sloppier terminology of "scaling violation" to emphasize the difference with the naive scaling expectation often assumed in the literature.

axions emitted by strings would be so large that nonlinear effects of the axion potential cannot be neglected, even considering the axion radiation in isolation, without topological defects.

We study the effects of these nonlinear dynamics in some detail. Their main consequence is a partial reduction of the number density of axions from strings, which however continues to dominate over the naive estimates based on the misalignment mechanism alone, or equivalently, over the results obtained by simulations of the full network of strings and domain walls carried out at the (currently available) unphysical values of the string thickness.

The article is structured as follows. We present our discussion of the most important points of the analysis and the key results in the main text, in Sections 2, 3 and 4. Meanwhile we give all the details of the various analyses, further studies, spin-off results, checks and generalization of the formulas, and interpretations in the Appendices. In particular, in Section 2 we present the results of simulations of the scaling dynamics and the axions produced by strings. In Section 3 we provide both analytical and numerical analysis of the effects of nonlinearities on the axion abundance from strings. In Section 4 we discuss the physical implications of the results and the assumptions and uncertainties behind them. In Appendix A we give details about the numerical simulations. In Appendix B we provide additional analysis of the properties of the string network during the scaling regime, including studies of string velocities, the decoupling of the heavy modes, the axion and radial mode spectra, as well as the systematics. In Appendix C we discuss how logarithmic effects are also visible in the dynamics of single loops in isolation. In Appendix D we identify when and how the scaling regime ends as the axion potential turns on. In Appendix E we give more details and results of both the analytical and numerical analysis of the nonlinear regime during the QCD crossover. In Appendix F we study the effects of the presence of topological defects during the QCD crossover on the evolution of the axion radiation produced during the scaling regime. Finally, in Appendix G we comment on the compatibility of our results with the existing literature.

## 2 Axions from Strings: The Scaling Regime

When the PQ symmetry is broken a network of axion strings forms [14–16] and this rapidly approaches an attractor solution [17–20] during the subsequent evolution of the Universe (extensive evidence for this was given in ref. [7]). The attractor is independent of the network's initial properties, allowing predictions to be made that are independent of the details of the PQ breaking phase transition and of the very early history of the Universe (i.e. at times much earlier than that of the QCD crossover).

The dynamics of the string network is highly nonlinear, and while models have been proposed to describe the main features of the attractor [9–13] they typically rely on a series of (unproven) assumptions. Instead we study the properties of the string network using numerical simulations. In these we integrate the classical equation of motion of the complex scalar field $\phi$ that gives rise to the axion numerically, assuming a radiation dominated Universe.[2] For simplicity we choose the Lagrangian

$$\mathcal{L} = |\partial_\mu \phi|^2 - \frac{m_r^2}{2v^2} \left( |\phi|^2 - \frac{v^2}{2} \right)^2 ,  \tag{1}$$

which leads to spontaneous PQ symmetry breaking at the scale $v$. The axion field $a(x)$ is related to the phase of the complex scalar field as $\phi(x) = \frac{v+r(x)}{\sqrt{2}} e^{ia(x)/v}$, while the radial mode $r(x)$ is a heavier field of mass $m_r$ associated to the restoration of the PQ phase.

---

[2]Given the attractor nature of the string evolution and the fact that the main axion contribution is produced just before the axion potential becomes relevant, we only assume that radiation domination starts at least before the QCD crossover transition.

The scale $v$ can be trivially reabsorbed in a rescaling of $\phi$, while the scale $m_r$ provides the normalization of the physical space and time scales over which the dynamics unfolds. While the clock of the UV physics associated to the radial mode ticks with intervals set by $1/m_r$, the more phenomenologically relevant clock associated to the IR axion physics ticks at a much slower pace set by the scale $1/H$, which keeps slowing down as the Universe expands. For this reason it is more useful to study the dynamics in terms of Hubble $e$-foldings $\log(m_r/H) = \log(t/t_0)$ ("log" for short), where $t$ is the Friedmann-Robertson-Walker time (with metric $ds^2 = dt^2 - R^2(t)dx^2$, $R(t) \propto t^{1/2}$) and $t_0$ is the reference time at $m_r = H$. With appropriate random initial conditions, strings automatically form in simulations and their dynamics are fully captured. Regions of space containing string cores are identified from the variation of the axion field around small loops of lattice points. Further details on our implementation and algorithms are given in Appendix A.

Limits on computational power allow us to evolve grid sizes of at most $4500^3$ lattice points. Meanwhile, the lattice spacing $\Delta$ should be such that $m_r\Delta \lesssim 1$ and the physical length of the box $L$ such that $LH \gtrsim 1$ to avoid introducing significant systematic uncertainties. As a result, simulations can only access relative small values of $\log(m_r/H) \approx \log(L/\Delta) \lesssim 8$. In contrast, the vast majority of the axions present when the axion mass becomes cosmologically relevant come from the emission of the string network in the prior few Hubble times. This happens shortly before the time of the QCD crossover when $\log(m_r/H) \simeq 60 \div 70$. Therefore properties of the string network measured in simulations must be extrapolated if reliable, physically relevant, predictions are to be obtained.

A simulation trick, often used in the literature, is to make $m_r$ vary with time as $m_r(t) = m_r(t_0)\sqrt{t_0/t}$ (the so-called "fat-string" trick). In this way the string thickness $1/m_r(t)$ stays constant in comoving coordinates. The maximum $\log(m_r(t)/H) = 1/2\log(t/t_0)$ that can be simulated remains unchanged, but this is reached over a much longer physical time, which allows far better convergence to the attractor regime. Although the simulations performed with this trick might lead to different quantitative answers, it is expected (and so far confirmed) that the qualitative behavior is the same. We performed most simulations with both $m_r$ constant ("physical") and with the "fat" trick. While we only use the data from the physical simulations to extract the relevant parameters, the results obtained with the fat trick make some features of the attractor solution more manifest and our interpretation of the string dynamics more robust.

## 2.1 String Density

The energy density stored in the string network can be written as

$$\rho_s(t) = \xi\frac{\mu_{\text{eff}}}{t^2}, \tag{2}$$

where $\xi$, the number of strings per Hubble patch, counts the total length $\ell$ of the strings inside a Hubble volume in units of Hubble length, namely $\xi \equiv \lim_{L\to\infty} \ell(L)\,t^2/L^3$, while $\mu_{\text{eff}}$ represents the effective tension of the strings, i.e. their energy per unit length. At late times, the latter is approximately equal to the tension of a long straight string in one Hubble patch $\mu = \pi v^2 \log(m_r/H)$, where $v$ is the PQ breaking scale, which we take equal to the QCD axion decay constant $f_a$ from now on (we will discuss how to adapt our results to the more general case $v = Nf_a$ in Section 3.3). Such an approximation captures $\rho_s$'s leading dependence on $H$ and the UV parameters of the theory ($f_a$ and $m_r$). Corrections from the boost factors and the curvature of the strings are discussed in Appendix B.3.

The dynamics of strings are well known to be logarithmically sensitive to the evolving scale ratio $m_r/H$. As mentioned above, the string tension is itself a linear function of this logarithm, and consequently the effective coupling of large wavelength axions with long strings scales

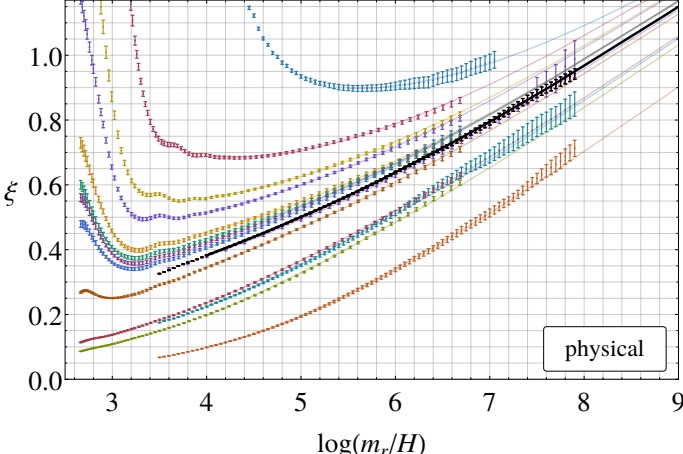

Figure 1: The evolution of the string network density $\xi$ for different initial conditions, with statistical error bars. Different initial conditions tend asymptotically to a common attractor solution. This has an evident logarithmic increase, which would imply $\xi \approx 15$ at the physically relevant $\log(m_r/H) = 60 \div 70$. The best fit curves with the ansatz in eq. (3) are also shown. The initial conditions used for the analysis of the spectrum of axions emitted by the network are plotted in black.

as $1/\log(m_r/H)$ (see e.g. ref. [21]). It is therefore not surprising that the dynamics of the string network, and in particular the parameters of the attractor, might depend non trivially on $\log(m_r/H)$. This is indeed the case for the parameter $\xi$, which was observed to "run" in ref. [7] (see also refs. [22–27] for further supporting evidence), increasing logarithmically with time.

The growth of $\xi$ is manifest in Fig. 1, which shows $\xi$ as a function of $\log(m_r/H)$. Each color refers to a set of simulations with different initial string density (initially overdense simulations show first a drop and then a universal increase). The error bars refer to the statistical errors.[3] Simulations ending before $\log = 7$ are data taken in ref. [7] with grids up to $1250^3$, and the remainder are new data collected with bigger grids, up to $4500^3$. When we analyze other properties of the scaling solution we choose the initial conditions that reach the attractor behavior the earliest, indicated with black data points in Fig. 1.[4]

Because of the manifest logarithmic increase, the value of $\xi$ at late times could be much larger than that measured directly in simulations. In ref. [7] it was shown that the data is compatible with a linear logarithmic growth. Here we extend that analysis including all the data sets with different initial conditions and with bigger grids, in total comprising about 1000 simulations of which 100 are with grids larger than $4000^3$. We test the linear logarithmic increase with the following fit ansatz (see Appendix B.1 for more details):

$$\xi = c_1 \log + c_0 + \frac{c_{-1}}{\log} + \frac{c_{-2}}{\log^2}, \tag{3}$$

where the coefficients $c_{-1,-2}$ are taken with different values for each data set to account for differing initial conditions, while the coefficients $c_{1,0}$, which survive in the large log limit, are taken universal across all data sets. As explained in [7] the string network starts showing scaling behaviors after $\log = 4$ (when strings can begin efficiently emitting axions with sub-horizon wavelengths), which we choose as our starting point for the fit.[5]

---

[3]These take into account both the total number of simulations and the number of independent Hubble patches in each simulation. For this reason the error bars increase toward the end of simulations where fewer Hubble patches are available.

[4]These are roughly those with the least overdense initial conditions.

[5]In order to avoid artificial bias in favor of data with higher frequency time sampling in the fit, we sampled

The result of the fit is represented by the colored curves in Fig. 1. The ansatz in eq. (3) reproduces all the data for a variety of initial conditions very well over almost 4 $e$-foldings in time. The $\mathcal{O}(1/\log)$ corrections are relevant only at the smallest values of the logs in the fit, while they become almost irrelevant by the end of the simulations.

The fit value of the slope $c_1 = 0.24(2)$ is definitely nonzero, confirming a non-vanishing universal increase. A straight extrapolation to $\log = 70$ would give $\xi = 15(2)$. The current precision however does not allow us to exclude an even steeper growth. In fact, a fit with an extra quadratic term (i.e. $c_2 \log^2$) gives analogously good results with a positive quadratic coefficient $c_2$, which would lead to even bigger values of $\xi$ at large logs. Simulations with the fat trick, which had more time to converge to the attractor, show an even more manifest linear log growth (see Appendix B.1). In particular the data set with initial conditions that reached the attractor the earliest in Fig. 6 leaves very little room for any nonlinear function to be a good fit. This suggests that $\xi$ has a linear behavior in both the physical and fat systems, as opposed to a steeper growth.

Because of the decoupling of the axion field at large values of the log, continued growth of $\xi$ beyond the reach of simulations would be compatible with the expectation that the global string network tends to approximate the Nambu–Goto string one (and the local string one) in the limit $\log \to \infty$. Indeed, old Nambu–Goto simulations gave values of $\xi_{\text{NG}}$ between 10 and 20 [18, 19, 28], while more recent local string ones [23, 29] give $\xi_{\text{loc}} = 4(1)$. This is a hint that $\xi$ for the global string network will not saturate at least prior to $\log \sim 20$ (extrapolating the linear growth).

An enhanced value of $\xi$ was also observed in global string networks in refs. [23, 27] where a large value of the effective string tension was achieved by means of a clever modification of the physics at the string core $m_r$.

However, we should point out that the asymptotic evolution of the string network parameter $\xi$ for axion strings has not yet been fully established. It is still unknown whether the decoupling of the axion from the string dynamics really completes within a finite range of logs or keeps going with an infinite running. As we will see further below, the axion spectrum extracted from field theoretic simulations still shows nontrivial changes in the dynamics that could qualitatively affect the asymptotic behavior of the network. On the other hand, Nambu–Goto simulations could also miss the asymptotic behavior of the network, as they lack the back-reaction of the bulk fields and Kalb-Ramond effective descriptions might not capture the physics of string reconnections and backreaction of UV modes properly. In fact even for local string networks, which are expected to already be in the Nambu–Goto limit, a nontrivial logarithmic evolution of $\xi$ might be present [23, 30].

To summarize, while we cannot exclude the possibility that the observed growth of $\xi$ saturates at larger values of the log, no indication of this is observed in the simulated range (it is particularly clear that the data for the fat system is incompatible with any reasonable function that plateaus soon after $\log = 8$), which suggests that such a saturation could potentially happen only at much later times, if at all.[6] Instead, all approaches seem to agree on a growth of $\xi$ to the range $\mathcal{O}(10)$ for $\log \sim \mathcal{O}(100)$, which is probably the most plausible and safe extrapolation. For our purposes we will assume the nominal value from our fit $\xi = 15$ for $\log = 60 \div 70$, taking into account that this estimate might receive $\mathcal{O}(1)$ corrections.

Another quantity that characterizes the string network is the distribution of string velocities. We study this property in Appendix B.2 where we show that, in agreement with other studies [22, 24, 31], the strings are mildly relativistic with an average boost factor $\langle \gamma \rangle \sim 1.3 \div 1.4$.

---

equally all simulation data taking one data point every time one Hubble patch reentered the horizon (and in doing so, correlations between data from the same simulation were also reduced). The most overdense set reaches the attractor later and has been fitted from $\log = 5.5$.

[6]Moreover the equations of motion contain no additional mass scales, which would break the self-similarity of the attractor solution, suggesting that the increase is likely to continue.

While this value appears to be approximately constant over the simulation time, the distribution of velocities shows a nontrivial evolution, with a subleading portion of the string network reaching increasingly higher boost factors as the log increases. This property is also compatible with the interpretation that the system is evolving towards the Nambu–Goto string network behavior, for which the formation of kinks and cusps explores arbitrary high boosts, and loops oscillate many times instead of shrinking and disappearing after one oscillation (more details are given in Appendices B.2 and C). As a consequence of the increasing Lorentz contraction from higher boosts, finite lattice spacing effects become more severe at larger values of the log. Such effects can be seen in a variety of observables (in particular in those that are more UV sensitive, see Appendix B for examples), and decrease the potential dynamical gain from simulations with bigger grids.

## 2.2 Axion Spectrum

In an expanding universe, eq. (2) and the conservation of energy imply that the string network continuously releases energy at a rate $\Gamma \simeq \xi\mu_{\text{eff}}/t^3 \simeq 8\pi H^3 f_a^2 \xi \log(m_r/H)$ (see e.g. [7] for more details). As shown in ref. [7], although most of this energy is emitted into axions, in simulations a non-negligible portion goes into radial modes (between 10% and 20%). Thanks to our new data with larger final logs, and by analyzing the radial excitation spectrum, we find that a significant part of the energy in radial modes is actually produced at the time the network enters the scaling regime and the subsequent emission of radial modes becomes less and less important (see the discussion in the Appendix B.4). This is compatible with the expectation that UV modes decouple from the network evolution at large values of the log (see Appendix B.3).[7] We will therefore assume that at late times the emission of radial modes is negligible and all the energy is released into axions.

The total energy density in axion radiation at late times is therefore $\rho_a \simeq 4/3\mu_{\text{eff}}\xi H^2 \log$, where the last log factor arises from the convolution of the emission rate over time.[8] As explained at length in ref. [7], the contribution of such radiation to the final axion abundance strongly depends on how the energy is distributed over axions of different momenta. A particularly useful quantity is the normalized instantaneous spectrum $F(k/H) = \partial \log(\Gamma)/\partial(k/H)$, which tracks the momentum distribution of axions produced at each moment in time by the string network. As mentioned in the Introduction, $F$ is expected to be approximately a single power law $F \sim 1/k^q$ between the IR scale set by Hubble and the UV one set by the string core. Depending on whether the spectral index $q$ is greater or smaller than unity, most of the axion energy density emitted is thus contained either in a large number of soft axions or in a smaller number of hard ones, with obvious implications for the resulting number density. For example, if $F$ is single power law $1/k^q$ with compact support $k \in [x_0 H, m_r/2]$, the axion number density turns out to be $n_a = 8\mu_{\text{eff}}\xi H \nu(q)/x_0$ where the function $\nu(q)$ rapidly interpolates between $1 - 1/q$ for $q > 1$ and $(H/m_r)^{1-q}$ for $q < 1$. It is therefore clear that the spectral index $q$ plays a crucial role in the determination of the axion abundance produced by the string network.

We extract $q$ from simulations using both the physical theory and the fat string trick, with the latter having a cleaner final spectrum with less residual dependence on the initial conditions. We fit $q$ in the range $30 < k/H < m_r/4$ over which it indeed shows a constant power law behavior. The fitting interval has been chosen somewhat smaller than that over which the network emits axions in order to further reduce possible systematics from finite volume and grid size effects. In Appendix B.4 we show that our results remain consistent as this range is changed, we discuss more properties of the spectra and give details of the simulations used.

---

[7]This also ensures that the dynamics of the string network are independent of the particular UV completion of the axion theory chosen in eq. (1).

[8]This expression for $\rho_a$ assumes radiation domination, and, in the large log limit, holds for any $\xi$ that has at most a logarithmic time dependence.

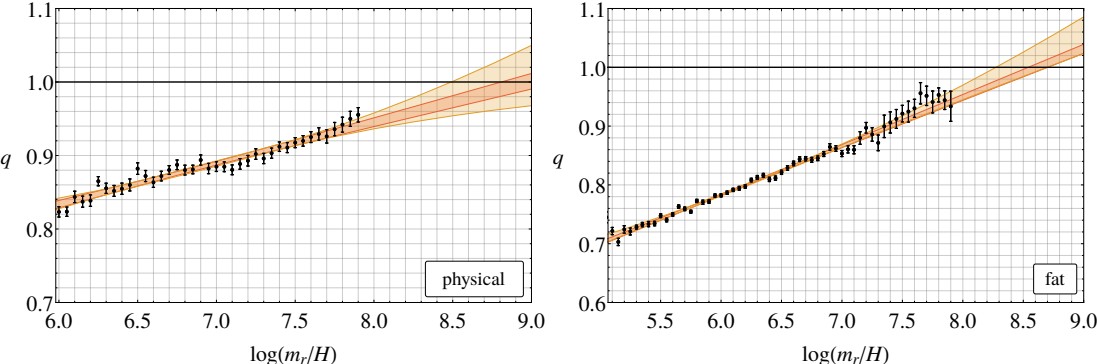

Figure 2: The spectral index of the instantaneous axion emission $q$ as a function of time (represented by $\log(m_r/H)$) for the physical (left) and fat string systems (right). The darker/lighter shaded regions correspond to the results of linear/quadratic fits to the data of the simulations (in black). The clear increase in $q$ implies that the axion spectrum is turning IR dominated (i.e. $q > 1$), a regime that it will reach long before the physically relevant $\log(m_r/H) = 60 \div 70$.

The value of $q$ as a function of $\log(m_r/H)$ is shown in Fig. 2. The data points represent the average of $q$ over many simulations and the error bars measure the associated statistical errors.[9] Although the spectral index is less than unity over the whole simulated range, a nontrivial growth is evident, corresponding to a spectrum that is becoming more IR dominated. The behavior is fit well by a linear function (i.e. $q(\log) = q_0 + q_1 \log$) in both the fat and the physical systems (the dark shaded region in Fig. 2). Fits with an extra quadratic term ($+q_2 \log^2$) give compatible results (the lighter shaded region in Fig. 2), although with larger uncertainties. This implies that the linear logarithmic growth will continue for, at the very least, a few more $e$-foldings.

Hence the data in Fig. 2 strongly suggests that the spectrum becomes IR dominated ($q > 1$) within one or two $e$-foldings beyond the simulation reach.[10] Note however that the data shown in Fig. 2 represent averages over many simulations: while at early times ($\log \lesssim 6$) all the simulations that comprise our data sets have $q < 1$, at late times ($\log \gtrsim 7.5$) a portion already shows an IR dominated instantaneous spectrum with $q > 1$. This strengthens our confidence that the spectrum indeed turns IR dominated at slightly larger values of log. Further suggestive evidence can be found in Figs. 14 and 15 of Appendix B.4.1, in which the shape of the instantaneous spectrum $F$ at different times is plotted.

This nontrivial log dependence of the emitted axion spectrum correlates with all the other evidence of evolution of the attractor's parameters, in particular with the reduction of UV mode emission. The most conservative extrapolation of the data in Fig. 2 is to values of $q$ larger than unity at late times. Fortunately, as we will explain in the next Section, as long as $q > 1$ the final axion abundance only has a very weak dependence on its precise value. For this reason we will not attempt to perform a real extrapolation of $q$ from the data in Fig. 2, but we will just assume that at $\log > 60$ its value is definitely larger than unity (say, $q > 2$).

To summarize, we performed dedicated high-statistics large-grid simulations of the axion string network, providing strong evidence for nontrivial evolution of the network's scaling parameters towards the expected behavior of Nambu–Goto-like strings. In particular, both the

---

[9]At late times the statistical errors increase because of the reduction in the number of independent Hubble patches in a simulation box. Meanwhile, at small values of the log the reduced range in the spectrum to fit $q$ (which is particularly important for physical simulations where the contamination from not-yet-fully-redshifted UV modes is more severe) counteracts the large number of Hubble patches available at these times.

[10]Confirming this directly would require grids of order $20000^3$ or bigger, which are beyond our current reach (but may be reachable in the coming years), or through improved numerical algorithms [32].

string density and the axion spectrum vary in a way that, once extrapolated to the physical parameter region relevant for QCD axions, can make the relic axion component produced by strings orders of magnitude larger than the naive one inferred directly from simulations.

The possibility that topological defects, in particular strings, might provide the dominant contribution of relic axions (much larger than the naive misalignment one) was already argued long ago [33–36], by assuming that at late time the axion string network's dynamics was well approximated by the Nambu–Goto one, and in particular $q > 1$. Our results in this Section represent the first clear evidence from full field theory simulations in support of this picture and provide a more detailed characterization of how this limit is approached.

## 3 From Strings to Freedom

The scaling regime discussed in the previous Section ends at temperatures of order the QCD scale, when the axion potential becomes relevant and the PQ symmetry is explicitly broken. At this time each string develops $N$ domain walls (where $N = v/f_a$ is the QCD anomaly coefficient). For $N = 1$ (we will discuss the case $N > 1$ in Section 3.3) there are no conserved quantum numbers left, and the network of strings and walls subsequently decays into axions.

As mentioned in the Introduction, a huge hierarchy of scales forbids a direct numerical study of the system at these times. Given the observed evolution of the properties of the string network (which dramatically changes the dynamics at large scale separations already during the scaling regime) we cannot trust results for the string/wall system dynamics from simulations that are carried out so far away from the physical point. Instead, we focus solely on the contribution of axions produced before the axion potential becomes relevant (i.e. on axions emitted while the system was still in the scaling regime), which requires far fewer theoretical assumptions and extrapolations. To do so, we will study the nonlinear evolution of these axions through the QCD transition in isolation, ignoring the presence of strings and walls and the additional axions they decay into. This allows us to perform direct numerical simulations without the need for any extrapolations. The price to pay is that we miss the component of axions that is produced from the decay of strings and domain walls, which will presumably contribute further to the abundance. In this way we obtain only a lower bound on the final abundance. One may worry that the strings and walls, and the axions produced from them afterwards, could interfere with the evolution of the preexisting axions that we are trying to reconstruct. However, barring an unlikely highly-efficient absorption of background axions by topological defects, their presence is not expected to alter our lower bound considerably, and at worst might weaken it by an order one factor (which, in any case, is not more than other sources of uncertainties that we will discuss at the end of the Section and in Section 4). This fact is further supported by a study in Appendix F.2 where we performed dedicated simulations to analyze the evolution of the axion radiation (as predicted by the scaling regime at $\log \sim 60 \div 70$) when strings and domain walls are included.

Away from topological defects the Hamiltonian density describing the propagation of the axion field is

$$\mathcal{H} = \frac{1}{2}\dot{a}^2 + \frac{1}{2}(\nabla a)^2 + m_a^2(t)f_a^2[1 - \cos(a/f_a)] , \qquad (4)$$

where, as suggested by the dilute instanton gas approximation [37] and supported by recent lattice simulations [38–42] (see also ref. [43] for a recent review), we assume that the axion potential at early times is described by a single cosine potential and the axion mass has a power dependence on the temperature $m_a \propto T^{-\alpha/2} \propto t^{\alpha/4}$, with $\alpha \simeq 8$ the preferred value.[11]

---

[11]The temperature dependence and the form of the axion potential is expected to change at $T \sim T_c \simeq 155$ MeV and below, where the axion potential is well approximated by the zero temperature prediction [44]. However, we

Naively one might think that the axions produced by strings propagate freely like radiation until their momenta (which is typically of order a few $H$) become of the same order as the axion mass $m_a$, after which they would start propagating as nonrelativistic matter. Throughout this whole process the comoving number density would be conserved. This is true if the axions remain weakly coupled for the whole time. Indeed the axion couplings are suppressed by either $H/f_a$ or $m_a/f_a$, most of the axions have small momenta of order $H \ll f_a$ and the effective coupling to strings is also suppressed by $1/\log \ll 1$.

However, as we will see below, the large quantity of axion radiation produced during the scaling regime implies that the average value of the field $\langle a^2 \rangle^{1/2} \gg f_a$, and nonlinear effects have an important effect on the axion number density. This whole process can be studied directly through numerical simulations of the axion field alone. The initial conditions are taken from the axion spectrum emitted by strings during the scaling regime extrapolated to the time $t = t_\star$, which we define as the moment when $H(t_\star) = m_a(t_\star)$, since we find that the axion spectrum is still unaffected by the potential at this point (see Appendix D). More axions will be emitted afterwards, however, since their spectrum is unknown, we conservatively do not include them in the initial conditions, and therefore not in our lower bound. Before presenting the results of our simulations we first describe what our expectations are for the effects of nonlinearities. In particular, we derive an analytic formula for the final axion abundance that agrees surprisingly well with the numerical results and correctly reproduces the dependence on the relevant parameters.

## 3.1 Analytic Description

As mentioned in Section 2.2, the energy density of the axions produced by the string network up until $t = t_\star$ is $\rho_a(t_\star) \approx \xi_\star \mu_{\mathrm{eff},\star} H_\star^2 \log_\star$ (from now on the subscript "$\star$" on a quantity indicates that it is computed at $t = t_\star$, $\log_\star \equiv \log(m_r/H_\star)$), where the last $\log_\star$ factor arises from the convolution of axion energy densities emitted over the course of the scaling regime. Using the results of Section 2 on the evolution of the network (in particular the fact that $q > 1$ long before $t_*$) the overall energy density $\rho_{a,\star}$ is distributed with a scale invariant spectrum (up to logarithmic corrections), i.e. $\partial \rho_a / \partial k \propto 1/k$, between the IR cutoff at $k \sim k_{\mathrm{IR}} = x_0 H_\star$ (with $x_0 = \mathcal{O}(10)$) and the redshifted UV scale at $k \sim \sqrt{H_\star m_r}$. We refer to Appendix E for the derivation of this result, and to eq. (23) for the explicit form of $\partial \rho_a / \partial k$.

The evolution of high frequency modes with $k \gg k_{\mathrm{IR}}$ is dominated by the gradient term even long after $t = t_\star$. Therefore, the nonlinearities arising from the axion potential are negligible for the entire evolution of these modes. As a result, we have to focus only on the IR part of the spectrum, the contribution of which to the energy density is $\rho_{\mathrm{IR}} \approx 8\xi_\star \mu_{\mathrm{eff},\star} H_\star^2 \sim 8\pi \xi_\star \log_\star H_\star^2 f_a^2$ (more precisely we define $\rho_{\mathrm{IR}}$ as the integral of the axion spectrum over momenta $k < c_m m_a$, with $c_m = \mathcal{O}(1)$ coefficient, since for higher modes the potential term is subleading). Given the extrapolated values of $\xi_\star$ and $\log_\star$ from Section 2, at $t_\star$ the IR axion energy density $\rho_{\mathrm{IR}} \sim \mathcal{O}(10^4) H_\star^2 f_a^2$ is much larger than the contribution from the axion potential ($\rho_V = m_a^2 f_a^2 [1 - \cos(a/f_a)]$), which is bounded by $\rho_{V,\star} < 2H_\star^2 f_a^2$. This means that at $t = t_\star$ most of the energy density is still contained in the gradient part of the Hamiltonian ($\frac{1}{2}\dot{a}^2 + \frac{1}{2}(\nabla a)^2$). Several implications follow from this fact.

First, since the gradient term dominates the Hamiltonian evolution of the field, even the modes with $k < m_a$, which in the linear regime would behave nonrelativistically, will not feel the presence of the potential term and so continue evolving as a free relativistic field after $t_\star$, until $\rho_V$ becomes comparable to $\rho_{\mathrm{IR}}$.

Moreover, since the typical gradient of the field is set by $H_\star$, in order for the gradient term

---

will see that for the range of parameters relevant for the QCD axion dark matter, the evolution of the axion field will turn linear at higher temperatures while the above ansatz is expected to still hold.

of the Hamiltonian density to account for $\rho_{\mathrm{IR}}$ the amplitude of the IR modes needs to be much larger than $f_a$, i.e. $\langle a^2 \rangle / f_a^2 \sim \mathcal{O}(\xi_\star \log_\star)$.[12] This means that at large $\xi_\star \log_\star$ the axion field is mostly a superposition of waves, with wavelengths of order Hubble, that wind and unwind the fundamental axion domain $(-\pi f_a, \pi f_a)$ several times in a topologically trivial way. Points in space with $a/f_a \sim \pi \mod 2\pi$ correspond to the core of domain walls with the topology of a sphere. For $\xi_\star \log_\star \gg 1$ there will be multiple domain walls nested inside each others, with a deformed onion-like structure. The presence of these domain walls however does not play any role as long as $\rho_V < \rho_{\mathrm{IR}}$ since the field continues to evolve freely.

During this period the field keeps redshifting relativistically, the amplitude of the field decreases, $\rho_{\mathrm{IR}} = \rho_{\mathrm{IR},\star}(t_\star/t)^2$ and the comoving number density of axions remains constant. Meanwhile, as the temperature continues to drop, approaching the QCD transition, the axion mass and $\rho_V$ increase rapidly. Eventually, at $t = t_\ell$ defined as the time when $\rho_{\mathrm{IR}}(t_\ell) = c_V \rho_V(t_\ell)$ (with $c_V$ an order one constant), the presence of the axion potential becomes important and the dynamics turn completely nonlinear. This corresponds to the moment when the domain walls start to be resolved, i.e. when the thickness of each domain wall ($\sim 1/m_a(t_\ell)$) has shrunk below the average distance between two walls ($\sim k^{-1} f_a/\langle a^2 \rangle^{1/2} \sim f_a/\rho_{\mathrm{IR}}^{1/2}(t_\ell)$). Soon after, domain walls, being topologically trivial, annihilate into axions. Except for few loci where oscillons can potentially form (which anyway can only take away a negligible portion of the total energy density, as shown in the next Section), the axion field amplitude continues to drop, rapidly falling below $f_a$. Nonlinearities fade away and conservation of the comoving axion number density is restored.

We can thus assume that during the nonlinear transient (at $t \sim t_\ell$) the axion energy density $\rho_{\mathrm{IR}} \sim \rho_V \sim m_a^2(t_\ell) f_a^2$ is promptly converted into nonrelativistic axions. The corresponding number density is $n_a^{\mathrm{str}}(t_\ell) = c_n \rho_{\mathrm{IR}}(t_\ell)/m_a(t_\ell) \simeq c_n m_a(t_\ell) f_a^2$, where $c_n$ is an order one coefficient taking into account transient effects, extra contributions from higher modes, etc. The value of $m_a(t_\ell)$ can be extracted from the definition of $t_\ell$ above and for $\alpha \gg 1$ (i.e. neglecting redshifting effects of $\rho_{\mathrm{IR}}$ with respect to the much faster axion mass growth) it is parametrically given by $m_a(t_\ell) \simeq (\xi_\star \log_\star)^{1/2} H_\star$. We therefore expect that, up to order one factors, the axion number density after the nonlinear regime is $n_a^{\mathrm{str}}(t_\ell) \simeq (\xi_\star \log_\star)^{1/2} H_\star f_a^2$, i.e. it is enhanced by a factor $(\xi_\star \log_\star)^{1/2}$ with respect to the misalignment contribution. Note that the enhancement, while substantial, is parametrically smaller than the naive one obtained by assuming that the axion field remains linear throughout the QCD transition, which would be $\xi_\star \log_\star$.

The main effects of the nonlinearities can be simply summarized as follows: the large energy density stored in the axion gradient term delays the moment when the axion mass and potential become relevant. In the meantime the axion mass is growing fast, so that, by the time the potential becomes relevant and the axions nonrelativistic, more energy is required to produce each axion and the comoving number density is suppressed.

The estimate above can be improved by keeping all the order one factors, taking into account the actual shape of the spectrum and the effects of redshifting from $t_\star$ to $t_\ell$. The full computation is discussed in Appendix E.1 and gives

$$ Q = \frac{n_a^{\mathrm{str}}(t_\ell)}{n_a^{\mathrm{mis},\theta_0=1}(t_\ell)} = A_{\xi_\star \log_\star, x_0} \left( \xi_\star \log_\star \right)^{\frac{1}{2} + \frac{1}{4+\alpha}}, \tag{5} $$

where $n_a^{\mathrm{mis},\theta_0=1}(t_\ell)$ is the axion number density from misalignment with $\theta_0 = 1$ redshifted to $t_\ell$, the prefactor $A_{\xi_\star \log_\star, x_0}$ is a function of all the parameters (including the order one coefficients $c_m, c_V, c_n$ but with only a mild logarithmic dependence on $\xi_\star \log_\star$, and $x_0$ (the full form is given in eq. (36)). The dependence on $q$ is further suppressed by $1/\log_\star$, as shown in Appendix E.1.

---

[12]See eq. (25) in Appendix E for an explicit derivation based on the spectrum.

The result in eq. (5) assumes that $\xi_\star \log_\star \gg 1$ and involve several approximations, parametrized by some unknown order one coefficients. These crudely describe the number of IR modes involved in the nonlinear dynamics ($c_m$), the relative importance of the potential versus the gradient energy in IR modes when nonlinearities become relevant ($c_V$) and the conversion factor of energy density into number density during the true nonlinear transient ($c_n$). While these numbers can only be fixed through numerical simulations, the full dependence on $\xi_\star \log_\star$ as well as the subleading ones on $\alpha$ and $x_0$ are genuine prediction of our analysis. As we will see next, they are nicely reproduced by the numerical simulations.

## 3.2 Comparison with Simulations

The dynamics discussed above can be checked by numerically integrating the axion equations of motion from the Hamiltonian density in eq. (4). We start the simulations at $t = t_\star$ with initial conditions set by the axion field radiation that would be produced during the scaling regime for different values of $\xi_\star \log_\star$ and $x_0$. The form of the spectrum is characterized by the position of the IR cutoff ($x_0$) and the spectral index of the instantaneous spectrum ($q$), while the overall size is controlled by the parameter $\xi_\star \log_\star$. We carry out simulations with different values of $\alpha$, which fixes the temperature dependence of the axion mass. More details are given in Appendix E.2.

For sufficiently large $\xi_\star \log_\star$ the numerical simulations show that the system indeed continues to evolve as in the absence of a potential after $t = t_\star$, redshifting as radiation and with a conserved comoving number density. More details and plots are given in Appendix E.3. The larger $\xi_\star \log_\star$ is, the longer the period of relativistic redshift lasts. This regime ends, as expected, with a nonlinear transient, after which the mean square field amplitude rapidly drops below $f_a$ (see Fig. 23).

At this point the field settles down around the minimum of its potential at $a = 0$, oscillating with an amplitude that is much smaller than $f_a$ almost everywhere. Consequently, the system becomes linear again except in a few localized regions of size $m_a^{-1}(t)$ where the field continues oscillating with an amplitude of the order $\pi f_a$. These objects, remnants of the large initial field amplitude (with $a > f_a$ at $t = t_\ell$), are known as oscillons or axitons [45, 46]. Oscillons are heavy and slowly decay radiating their energy density into axion waves with momentum of order $m_a$. Their lifetime is long enough that they persist until the end of our simulations. However, only a very small portion of the energy density remains trapped in oscillons, so that their presence is irrelevant for the computation of the final axion abundance. More details about the oscillons can be found in Appendix E.3.

Everywhere else the axion field is in the linear regime by the end of the simulations. We can therefore calculate the total axion spectrum $\partial \rho_a / \partial k$ and number density $n_a^{\text{str}} = \int dk (\partial \rho_a / \partial k) / \omega_k$ ($\omega_k = \sqrt{k^2 + m_a^2}$). We do so screening away the regions occupied by oscillons, and we use the difference with the unscreened results to estimate the uncertainty introduced by the presence of these objects. As anticipated the difference is small, which confirms that only a negligible portion of the energy density is trapped in oscillons. Moreover, as expected, after the screening the conservation of the comoving number density further improves. Additional discussion and plots are given in Appendix E.2. Thanks to the rapid growth of the axion mass, the nonlinear regime is reached not long after $t_\star$ and the system soon becomes linear again, after a short transient, as the field relaxes below $f_a$. For this reason, in the range of $\xi_\star \log_\star$ and $f_a$ under consideration ($\xi_\star \log_\star \lesssim 10^5$, $f_a \lesssim 10^{11}$ GeV), the system reenters the linear regime (and our simulations end) at temperatures that have dropped by, at most, a factor of four from that at $t_\star$. This is still above the QCD transition, in a regime where the axion potential used in eq. (4) should hold.

The agreement between the numerical simulations and our analytic description is not only qualitative but also quantitative. We compare the two using the ratio $Q = n_a^{\text{str}} / n_a^{\text{mis},\theta_0=1}$ of

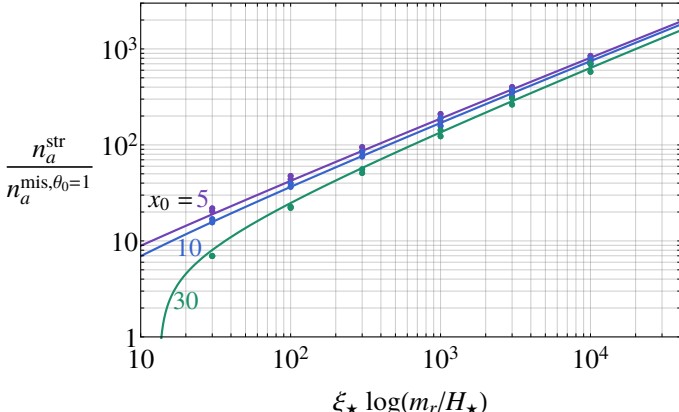

Figure 3: The late time ratio between the axion number density from strings and from misalignment (with $\theta_0 = 1$) as a function of $\xi_\star \log_\star$ for varying $x_0$ (the IR cutoff of the axion spectrum in units of Hubble) and fixed $\alpha = 8$ (the power law controlling the temperature-dependence of the axion mass). The data points are the results from simulations with statistical errors, and the curves correspond to the analytic prediction of eq. (5) (see Appendix E.1 and eq. (36) for more details).

eq. (5), between the number density of axions from strings and the reference one from misalignment (with initial misalignment angle $\theta_0 = 1$). Since both $n_a^{\mathrm{str}}$ and $n_a^{\mathrm{mis},\theta_0=1}$ are conserved per comoving volume at late times, $Q$ asymptotes to a constant value. The results for $Q$ for $\alpha = 8$ and $x_0 = 5, 10, 30$ are plotted in Fig. 3 as a function of $\xi_\star \log_\star$, and the comparisons for the other values of $\alpha$ are reported in Appendix E.2. The three parameters of the analytical formula $c_m, c_V, c_n$ have been fixed with a global fit of $Q$ including all the simulations with different values of $\alpha$, $x_0$ and $\xi_\star \log_\star$. The agreement between the theoretical estimate and the simulation data is remarkable given that: 1) all data is fit with just three universal parameters which indeed turn out to be of order one, and 2) the dependence on $\xi_\star \log_\star$, which is a prediction (not the result of a fit), agrees very well over multiple orders of magnitude. The only slight deviation is at low values of $\xi_\star \log_\star$ where the approximations used in the analytic formula are not in fact valid. More details about the dependence on the input spectrum, the values of the fitted input parameters and the dependence on the other parameters can be found in Appendix E.2. Here we simply note that, as anticipated, the dependence on the spectral index $q$ of the spectrum from the scaling regime is negligible as long as $q$ is away from unity. Although the numerical simulations are capable of covering the parameter space relevant for the QCD axion (discussed in Section 4), our analytic formula, in addition to providing a better understanding of the physics behind the nonlinear effects, would allow us to interpolate and extrapolate the simulation results to other values of the parameters if needed.

We will discuss the phenomenological implications of our results in Section 4; first we analyze the effects of nonlinearities on the shape of the final axion spectrum in more detail. As shown in Fig. 25 in Appendix E.3, the spectrum continues to redshift almost unaltered after $t_\star$ until it reaches the nonlinear regime at around $t = t_\ell$. At this point the energy contained in modes $k \lesssim m_a(t_\ell)$, is converted into massive nonrelativistic axions. For this to happen axions with $k \lesssim m_a(t_\ell)$ need to combine with each other to generate on-shell axions with mass $m_a(t_\ell)$, and the comoving number density of this component cannot be conserved. In other words, nonlinearities remove the IR part of the spectrum via 3-to-1, 5-to-3, etc. processes. The smaller $k$-modes are those with the larger occupation number and they therefore suffer stronger nonlinear effects. The resulting spectrum after the end of the nonlinear transient will therefore be peaked at physical momenta that were of order $m_a(t_\ell)$ at $t = t_\ell$, which is significantly higher than the would-be peak at $x_0 H_\star$ (at $t = t_\star$) had the nonlinearity been

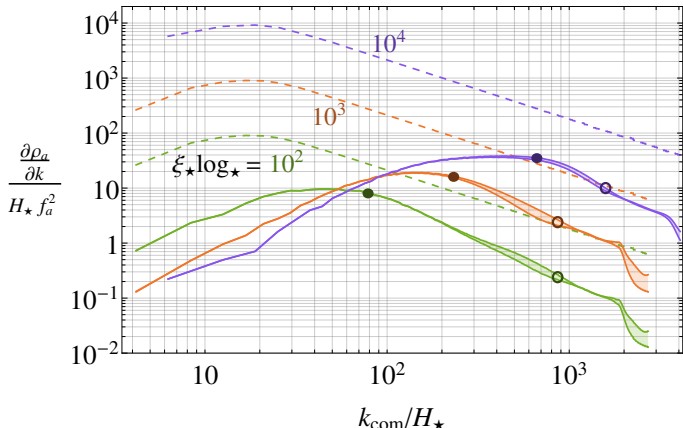

Figure 4: The axion energy density spectrum at $H = H_\star$ (dashed lines) and after the nonlinear transient (at the final simulation time $H = H_f$, solid lines) for $\xi_\star \log_\star = 10^2$ (green), $10^3$ (orange) and $10^4$ (purple), as a function of the comoving momentum $k_{\mathrm{com}} \equiv k(H_\star/H)^{1/2}$. The lower and upper (solid) lines of the same color correspond to the results obtained with and without the oscillons masked. The unfilled dots show the comoving momenta corresponding to the physical momenta that are equal to the axion mass at the final time (i.e. $k_{\mathrm{com}} = m_a(t_f)(H_\star/H_f)^{1/2}$). We also indicate the comoving momenta corresponding to the physical momenta that are equal to the axion mass at $t = t_\ell$ (i.e. $k_{\mathrm{com}} = m_a(t_\ell)(H_\star/H_\ell)^{1/2}$, filled dots), which is parametrically the time when the nonlinear transient occurs.

absent. In particular, the value at the peak grows with $\xi_\star \log_\star$. This is shown in Fig. 4 where we plot the spectrum as a function of the comoving momentum $k_{\mathrm{com}} \equiv k(H_\star/H)^{1/2}$ for the three values of $\xi_\star \log_\star = 10^2, 10^3, 10^4$, at the initial time $t = t_\star$ and at the final simulation time.

The deformation of the spectrum above could have important implications for the properties of the small scale structures produced by the axion inhomogeneities known as miniclusters [47] (see also [25, 26, 48, 49] for recent studies).

Figure 4 also shows the role of oscillons. These only affect the spectrum at momenta of order $m_a$, indicated by empty dots (see Appendix E.3 for details). Since the largest contribution to the number density comes from the peak of the spectrum, once $m_a(t)$ is sufficiently above this (as is the case at late enough times) the screening of oscillons does not significantly affect the measured axion number density. This matches the results for the number density evaluated directly, described above.

We finish this Section by briefly discussing the possible effects of the presence of strings and domain walls during the QCD transition, which have been omitted so far. We first note that at $t = t_\star$ the energy density in the string network is comparable to that we considered from the IR part of the axion radiation ($\rho_{\mathrm{IR}}$), and it is mostly localized along the strings themselves, so the dynamics of the field away from the strings should be largely unaffected by their presence. After $t_\star$ domain walls start to form but their energy density is bounded by the axion potential, and becomes relevant only much later, when the axion field has relaxed to values $a \sim \pi f_a$ everywhere. Hence away from strings we do not expect the dynamics of the axion field to be significantly different from those we computed, at least until $t \sim t_\ell$. At this point the nonlinear transient starts. The difference with respect to our simplified case is that, as well as our topologically trivial domain walls, extra walls surrounded by strings are also present. If the extra string-wall network decays during the transient, then as we saw before the total energy density (that in strings walls and radiation) is expected to convert into axions with a conserved comoving number density of order $(\rho_{tot}(t_\ell)/m_a(t_\ell))$. If for some reason[13] the

---

[13]One possibility could be that, analogously to string loops, which at large values of the log are expected to

string-wall network were to survive for longer, away from them the field would still evolve as calculated above. Therefore, we would expect that the results given above should represent, up to $\mathcal{O}(1)$ factors, a lower bound on the axion abundance regardless.

It would be quite surprising if the extra string-wall system were able to wipe away the bulk axions with a high enough efficiency to suppress their big contribution to the final abundance significantly. To further exclude this possibility we performed dedicated simulations where, as well as the axion radiation predicted by the scaling regime, we also included the strings (and the domain walls that form from them) during the mass turn on. In these simulations, the background axion radiation is as it would be with the physical parameters (i.e. with the spectrum and energy density expected at $\log_\star \sim 60 \div 70$). However, the string-domain wall network is evolved with the currently allowed $\log(m_r/H) \lesssim 7$, so $\xi_\star \log_\star$ for the string system is much smaller than the physically relevant value. As expected, the presence and decay of strings and domain walls does not significantly alter the evolution of the preexisting background radiation, and thus does not decrease the final abundance. Since in such simulations $\xi_\star \log_\star$ for the string network is small, and the emission from the decay of strings is UV dominated, the inclusion of strings also does not noticeably increase the final abundance. We refer to Appendix F for more details and the explicit results of these simulations.

From this study we learn two important lessons. Calculations of the axion abundance from brute force simulations of the whole evolution of the string-domain wall system can easily miss the dominant source of axion emission, underestimating the final relic abundance by more than one order of magnitude. Moreover, the explicit inclusion of strings in the late evolution of the field does not play a role unless their contribution starts becoming comparable to that from radiation during the scaling regime, at which point a tuned cancellation among the two sources would be surprising.

### 3.3 The case $N > 1$

We now discuss the generalization of our results to the case $N = v/f_a > 1$. First notice that in the equations of motion from the Lagrangian in eq. (1) the scale $v$ can be removed by rescaling the complex scalar field $\phi$. This means that the string dynamics during the scaling regime do not depend on $v$. The way $v$ enters observables is just fixed by dimensional analysis, and in particular all energy densities, number densities and the string tension are proportional to $v^2$. Therefore, the axion spectrum produced during the scaling regime in the general case can be recovered by simply multiplying the results of Section 2 by $N^2$.

On the other hand, the axion potential produced by QCD in eq. (4) involves the scale $f_a$. In all our computations in Section 3, the scale $v$ only enters through the axion spectrum via the scaling solution used as an input, where it appears in combination with $\xi_\star \log_\star$. All the results in Section 3 can therefore be generalized by simply substituting $\xi_\star \log_\star$ with $N^2 \xi_\star \log_\star$ (e.g. in eq. (5) and in Figs. 3 and 4).

The effect of $v > f_a$ is therefore to increase the energy density of the axions produced by strings (as a result of the enhanced string tension), increasing the field amplitude and therefore the effects of nonlinearities. Roughly, the final number density of axions will be enhanced by an $\mathcal{O}(N)$ factor, and the peak in the final spectrum will be UV shifted by a similar amount.

---

oscillate many times before shrinking and disappearing, domain wall disks surrounded by string loops might also behave similarly in this regime.

## 4 Results and Phenomenological Implications

We can now extract some phenomenological implications from the results of the previous Sections. In particular, given the extrapolated values of the axion spectrum from the string scaling regime, Fig. 3 and Section 3.3 provide a lower bound on the axion number density (in terms of the easily computed misalignment result). This can be translated into corresponding bounds on the axion mass and its decay constant requiring that such an abundance does not exceed the current observed dark matter value. As reference values we choose $\xi_\star = 15$, $x_0 = 10$, $q > 2$, $\alpha = 8$, $\log_\star = 64$, which for $N = 1$ (as in the minimal KSVZ model [50, 51]) imply

$$m_a \gtrsim 0.5 \text{ meV}, \qquad f_a \lesssim 10^{10} \text{ GeV}, \tag{6}$$

while for $N = 6$ (as occurs in e.g. the DSFZ model [52, 53]) they imply

$$m_a \gtrsim 3.5 \text{ meV}, \qquad f_a \lesssim 2 \cdot 10^9 \text{ GeV}. \tag{7}$$

For comparison, the naive axion number density from misalignment in the post-inflationary scenario is obtained by averaging the misalignment relic abundance with a flat $\theta_0$ distribution in the interval $[0, 2\pi]$. This gives $n_a^{\text{mis,avg}} \simeq 5 n_a^{\text{mis},\theta_0=1}$, which corresponds to $m_a \simeq 0.028$ meV and $f_a \simeq 2 \cdot 10^{11}$ GeV, more than an order of magnitude weaker than our bound.

We do not think that it would be fair to associate an error to the figures in eqs. (6) and (7): shifts of $\mathcal{O}(1)$ could be expected, but we would be surprised if these bounds relax by significantly more than a factor of two. To provide a better feeling for the main sources of uncertainty, and the choices of parameters used, we will now go through all the assumptions underlying the numbers above:

$\xi_\star$: We fixed the value of $\xi_\star = 15$ from the best fit of the scaling solution described in Section 2. As discussed at length this number assumes that the linear-log behavior observed in simulations extends beyond the simulation range by another order of magnitude.[14] While such an assumption can be questioned, other independent studies support a similar enhanced value. These include refs. [23, 27], which partially reproduce the possible effects of an enhanced string tension and find $\xi_\star \simeq 5$; Nambu–Goto simulations, which seem to prefer values between 10 and 20 [18, 19, 28]; and recent local string simulations, which give $\xi_{\text{loc}} = 4(1)$ [23, 29]. Since the final abundance approximately scales as $\xi_\star^{1/2}$, even assuming that the growth of $\xi$ saturates at the smaller values $\xi_\star \simeq 4 \div 5$, this would affect the final bound by less than a factor of 2, within our target precision. Substantially larger deviations from our central value seem unlikely.

$q$: The main assumption behind the result above is associated to the spectral index $q$ being larger than unity. Although present simulations cannot provide a proof, our analysis in Section 2 shows that $q > 1$ is by far the most conservative extrapolation of the results from simulations. This extrapolation is also supported by theoretical arguments about the expectation that the string network approaches the Nambu–Goto dynamics at large values of $\log_\star$. With $q > 1$ the instantaneous axion spectrum emitted by strings is IR dominated. The corresponding integrated spectrum (which determines the final abundance) is therefore fixed and only very weakly dependent on the actual value of $q$. As a result, the actual extrapolated value of $q$ does not lead to large uncertainties in our estimate (see Appendix E.3).

$\log_\star$: We set the reference value of $\log_\star = 64$, corresponding to fixing $f_a \simeq 10^{10}$ GeV, $m_r \simeq f_a$ and the value of $\alpha = 8$ (discussed below). Much smaller values for $\log_\star$ are in principle

---

[14]A similar linear-log increase has been seen in independent studies of global strings in [22–27].

allowed ($m_r \gtrsim$ keV from astrophysical and fifth force experimental constraints) although these are much less plausible given that the smallness of $m_r$ would not come for free.

$x_0$: We set the position of the IR cutoff of the spectrum $x_0 = 10$ from the results of the simulations at $\log \lesssim 8$ (see Fig. 14). Within the available range of simulations $x_0$ is consistent with being constant, although we cannot exclude a slow evolution which would change its value at large $\log_\star$. One might indeed expect that $x_0$ could increase with $\xi^{1/2}$, as the average interseparation between strings is reduced.[15] The study of the evolution of $x_0$ from simulations is more challenging than that of $q$ since it is much more sensitive to finite volume effects and it requires a better understanding and modelling of the shape of the IR peak. Fortunately, as discussed in the previous Section and explicitly shown in Appendix E.1, the final abundance is only logarithmically sensitive to $x_0$, so even a substantial change at large values of $\log_\star$ has a limited impact on the final result. This can also be seen in Fig. 3.

$\alpha$: We set the index that controls the temperature dependence of the axion potential to the value $\alpha = 8$, which corresponds to the prediction of the dilute instanton gas approximation at weak coupling. Although this computation is probably out of its regime of validity at the temperatures we are interested in, the same value of $\alpha$ seems to be supported by the most recent lattice QCD simulations. Waiting for an independent check we adopt this value as the reference one and provide the results for generic $\alpha$ in Appendix E. Similarly the results in refs. [38, 42] suggest that a simple cosine is a good approximation to the axion potential for the temperatures relevant to the nonlinear regime.[16]

– Extra strings-domain walls contribution: The last source of systematic error comes from neglecting the extra contributions from strings and domain walls present after $t_\star$. As discussed at length in the previous Section, we expect these to add further to the axion abundance, hence our lower bound. If the extra contribution is subdominant then our bounds would turn into central values for the abundance. If the extra contribution dominates, they will become strict inequalities. We cannot exclude a partial destructive interference between these extra contributions and the axion spectrum produced at earlier times. However it would be highly unlikely that it could weaken the bounds in eqs. (6) and (7) beyond an $\mathcal{O}(1)$ factor, i.e. by more than the size of the other uncertainties.

When combined with astrophysical constraints, the bounds in eqs. (6) and (7) restrict the allowed parameter space for the QCD axion in the post-inflationary scenario quite substantially. In particular, they motivate efforts to further explore a region of parameter space that could in principle be probed by astrophysics, as well as axion dark matter [54–57] and non dark matter [58–60] experiments. In Fig. 5 we show our bound for the QCD axion mass in the post-inflationary scenario, together with constraints on the axion-photon coupling $g_{a\gamma\gamma}$[17] from currently running experiments and the parameter space that could be probed by proposed experiments.[18]

---

[15]We thank Javier Redondo for a discussion on this point.

[16]We also note that the uncertainty introduced by the number of degrees of freedom in thermal equilibrium at $t = t_\star$ and $t = t_\ell$, and by the changes between these times, is relatively small, certainty within our target precision.

[17]Defined as $\mathcal{L} \supset -\frac{1}{4} g_{a\gamma\gamma} a F \tilde{F}$ in the low energy theory, where $F$ is the electromagnetic field strength.

[18]The existing experimental and observational bounds shown are from ADMX [61,62], earlier cavity experiments "UF/RBF" [63, 64], HAYSTACK [65], CAST [66], observations of horizontal branch starts "HB" [67], supernova 1987a "SN1987a" [68–70] (see however [71]) and red giants and white dwarf stars "RG/WD" [72–75]. The constraints on DSFZ axions from supernova 1987a, red giants and white dwarfs are model dependent via the mixing angle $\beta$. For those from white dwarfs and red giants we plot the limit from [75] for $\tan \beta = 1$. The limit from supernova 1987a is $m_a < 0.02$ eV for $\tan \beta = 1$ and this barely weakens for smaller $\tan \beta$ but it

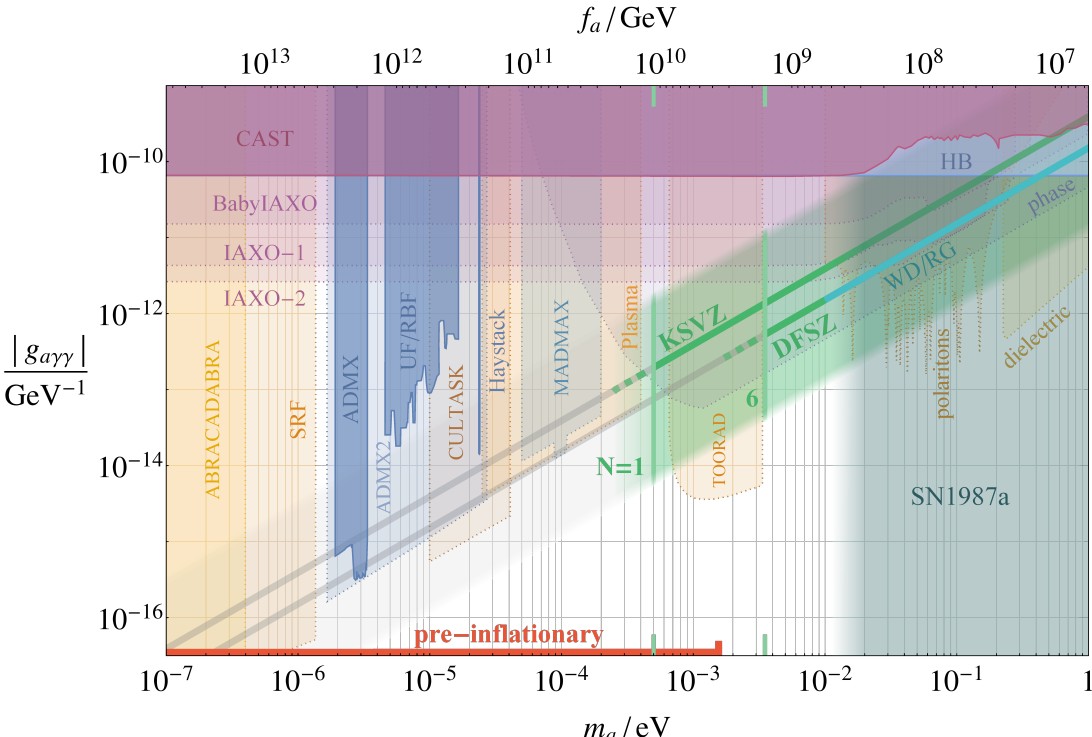

Figure 5: The axion parameter space in terms of its mass and coupling to photons $g_{a\gamma\gamma}$. Solid green lines indicate the allowed parameter space in the post-inflationary scenario for the minimal KSVZ model and the DFSZ model, given our constraints from dark matter overproduction (eqs. (6) and (7)), while the dashed green lines indicate our estimate of the uncertainty on these results. The green shaded band indicates the post-inflationary parameter space allowed by the bound for more general axion models. Vertical green lines show our lower bounds on the axion mass at $m_a = 0.5$ meV and $m_a = 3.5$ meV for $N = 1$ and $N = 6$ respectively. We also indicate, in red, the allowed axion masses in the pre-inflationary scenario (the corresponding $g_{a\gamma\gamma}$ lie in the partially transparent grey band), the upper limit of which $m_a \lesssim 1.5 \cdot 10^{-3}$ eV is set by isocurvature fluctuations. Existing experimental bounds and observational constraints (solid lines) on $g_{a\gamma\gamma}$ as a function of the axion mass and the projected sensitivity of proposed experiments (dotted) are also shown. The limit on DFSZ models from white dwarfs and red giants ("WD/RG") is indicated for $\tan\beta = 1$ [75], while the supernova-1987a limit on such models ("SN1987a") spans the blurred region as $\tan\beta$ varies (the corresponding constraint on KSVZ models is $m_a < 15$ meV) [70]. The post-inflationary region that we identify could also be probed by future experiments sensitive to the axion's couplings to matter [54, 59]. In combination with the bound from supernova, the region of viable QCD axion masses in the post-inflationary scenario is restricted to $m_a \approx 0.5 \div 20$ meV.

Interestingly, the allowed window is almost complementary to that of the pre-inflationary scenario. The upper bound on the mass in this case is $f_a \gtrsim 3.7 \cdot 10^9$ GeV and comes from requiring that the Hubble parameter during inflation is small enough to avoid observational constraints on isocurvature from Planck [83], but at the same time above $m_a$ so as not to deplete the misalignment abundance during inflation. In fact, in the overlapping region both the pre-inflationary and post-inflationary scenarios predict nontrivial small scale structures from the axion self-interactions [84], although the details are expected to differ as a consequence of the different origins of field inhomogeneities.

## Acknowledgements

We thank M. Buschmann, J. Foster, G. Moore, J. Redondo, K. Saikawa, B. Safdi and M. Yamaguchi for discussions. We thank CERN, GGI and MIAPP for hospitality during stages of this work. We acknowledge SISSA and ICTP for granting access at the Ulysses HPC Linux Cluster, and the HPC Collaboration Agreement between both SISSA and CINECA, and ICTP and CINECA, for granting access to the Marconi Skylake partition. We also acknowledge use of the University of Liverpool Barkla HPC cluster.

## A  The String Network on the Lattice

In this Appendix we summarize the methodology behind our numerical simulations of the scaling regime. In these we evolve the equations of motion of the Lagrangian in eq. (1),

$$\ddot{\phi} + 3H\dot{\phi} - \frac{\tilde{\nabla}^2 \phi}{R^2} + \phi \frac{m_r^2}{f_a^2}\left(|\phi|^2 - \frac{f_a^2}{2}\right) = 0\,, \tag{8}$$

where $\tilde{\nabla}$ is the gradient with respect to the comoving coordinates, on a discrete lattice with a finite time-step[19] (we fix $v = f_a$ as in Section 2). We assume radiation domination in a spatially flat Friedmann-Robertson-Walker background, so the scale factor grows as $R(t)/R(t_0) = (t/t_0)^{1/2}$, where $t_0$ is the initial simulation time and $H \equiv \dot{R}/R = 1/(2t)$. The comoving distance between lattice points remains constant, so the corresponding physical distance grows as $\Delta(t) = \Delta(t_0)(t/t_0)^{1/2}$.

We carry out simulations of the physical string system, for which $m_r$ is constant, and also the so-called fat string system in which $m_r(t) = m_r(t_0)(t_0/t)^{1/2}$. The core-size of strings is characterized by the length scale associated to the region where $|\phi| < f_a/\sqrt{2}$ and this is set by $m_r^{-1}$. Consequently, for physical strings the number of lattice points per string core decreases through a simulation, while for the fat string system it remains constant.

As discussed in Section 2, for a given grid size the maximum $\log(m_r/H)$ that a simulation can reach is limited by the simultaneous requirements that systematic errors from the finite lattice resolution and from the finite box size do not become too large. The former constrains the maximum value of $m_r \Delta$, while the latter imposes a lower bound on $HL$, where $L$ is the

---

strengthens slightly (up to the edge of the blurred region) for large $\tan\beta$ [70]. The proposed experiments shown are ABRACADABRA [76], superconducting radio frequency cavities "SRF" [77] (see also [78]), CULTASK [79], MADMAX: [80, 81], tunable plasma haloscopes "Plasma" [56], TOORAD [55], phase measurements in cavities "phase" [60], absorption by gapped polaritons "polaritons" [57] and IAXO [58].

[19]Many of the details of our implementation follow those described in Appendix A of [7]. For example, it is most convenient to work in terms of the rescaled field $\psi = R(t)\phi/f_a$, so that the Hubble term in eq. (8) is canceled. Rather than reviewing all such technicalities, here we focus on the key features and the differences in our present work.

physical box length, defined in Section 2. The corresponding maximum $\log(m_r/H)$ is the same for simulations of fat and physical strings, however the fat string system evolves for a longer cosmic time before this is reached. The maximum gridsize is limited by the available computational resources, and we carry out simulations with up to $4500^3$ lattice points.[20] The relatively large values of $\log(m_r/H)$ accessible with such grids are vital in identifying the evolution of $q$ described in the main text.

The numerical values of $m_r\Delta$ and $HL$ that can be used without introducing significant errors must be determined by direct testing in simulations. For $\log \lesssim 6.5$, $m_r\Delta = 1$ is sufficient for most observables of interest [7], however the bigger values of log in our present simulations necessitates that these are re-analyzed. We carry out a study of the finite lattice spacing effects from $m_r\Delta$ in Appendix B, where we show that some observables are indeed increasingly sensitive as log increases. To maximise the accessible value of log, we ran simulations until $HL = 1.5$. For this value $\xi$ still coincides with the infinite volume limit (see Appendix B.1). The IR part of the spectrum starts being slightly distorted, but not in the range of momenta used for the extraction of $q$, which still coincides with the infinite volume limit (see Appendix B.4.2). With these choices of $m_r\Delta$ and $HL$, our simulations reach $\log \simeq 8$.

## A.1 Selecting the Initial Conditions

For simulations to show the properties and log dependence of the attractor solution as clearly and accurately as possible, the initial conditions need to be fixed as close as possible to the scaling solution. If this is not done the network will go through a transient period as it approaches the attractor, decreasing the range of log over which its properties can be reliably studied. Indeed, the dynamics during the transient will differ from those in the scaling regime, and $\xi$ and $q$ might not show their true asymptotic evolution.

One requirement to be on scaling is connected to the initial density of strings. Simulations with too small a density will fail to reproduce the right properties associated to string interactions responsible for maintaining the attractor regime. Meanwhile, too large densities will lead to an enhanced string interaction rate and an overproduction of radiation with respect to the scaling regime. A possible criterion to identify the optimal initial conditions is to choose those with the highest density of strings that do not show a clear initial drop of $\xi$ before the observed universal asymptotic growth.

Another source of systematic noise is associated to initial excitations of the strings core. For example, such excitations will be triggered if the initial configuration contains strings with core-sizes that are significantly different to those on the attractor, which are parametrically set by $m_r^{-1}$. As the network evolves the strings cores relax to the properties they have on the attractor regime emitting UV radiation that pollutes the axion spectrum (mostly around the frequency $m_r/2$, as a result of the parametric resonance with the radial modes – see Appendix B.4). Although at late times ($\log \simeq 60 \div 70$) such radiation is completely negligible because of the huge redshift, the effect can be sizable in the limited extent of simulations.

The initial conditions are more important when studying the physical string system than the fat string one for two reasons. First, thanks to the longer cosmological time range, the fat string system reaches the attractor in a fraction of the total time (i.e. at smaller values of the log) even with untuned initial conditions, and the radiation left over from early times is diluted fairly efficiently by redshifting. In contrast, for physical strings the transient can easily last for the entire span of the simulation. Second, in the fat string system $m_r/2$ corresponds to a fixed comoving momentum. Therefore, the radial and axion modes emitted by the string cores only affect the UV part of the spectrum, outside the region of interest. Meanwhile, for physical strings these modes are redshifted towards smaller frequencies, polluting part of the

---

[20]To achieve this we use MPI parallelization across multiple (up to 48) cluster nodes.

spectrum used to extract the instantaneous emission $F$ by creating large oscillations (we will see this in more detail in Appendix B.4).

We use initial conditions that contain a fixed (adjustable) density of strings. For the fat string system, these are obtained by evolving eq. (8) starting from a random field configuration until the required total string length inside the box is reached. The field at this time is then used as the initial condition for the main simulation. The strings produced by such a procedure do not generally have the correct core size, since the Hubble parameter at the end of the initial simulation does not match that at the start of the main simulation. However, as mentioned above, the subsequent readjustment of the string cores only affect the UV part of the spectrum and has no consequences for the study of the attractor properties.

For simulations of the physical string system we modify this procedure slightly to overcome the issue with the spectrum discussed above. We generate initial conditions for these by evolving eq. (8) with $R \propto t$ and $m_r \propto R^{-1}$, until the desired total string length is reached. This choice has the advantage that both the Hubble parameter $H = \dot{R}/R$ and the string core size $m_r^{-1}$ are constant in comoving coordinates, i.e. $H^{-1}/R$ and $m_r^{-1}/R$ do not change. We chose $m_r^{-1}/R$ equal to $m_r^{-1}/R(t_0)$, i.e. the core-size at the beginning of the actual simulation. Consequently, the strings in the initial conditions have the right core-size, no matter what time the preliminary evolution ends. Moreover, the simulations to generate the initial conditions can run for an arbitrarily long time, since the comoving Hubble radius and the comoving core-size do not change. For small $H^{-1}/R$ this evolution corresponds to a system with large Hubble friction, which acts as a relaxation period smoothing out fluctuations of the initially random field and diluting preexisting radiation. Meanwhile strings form and their core-sizes relax. We choose $H^{-1}/R = H^{-1}(t_0)/R(t_0)$. With this value the total string length in the box decreases fairly slowly, so the relaxation period lasts a significant amount of time.

For the fat string system we start the main simulations at $\log(m_r(t_0)/H(t_0)) = 0$. For the physical system we found that cleaner initial conditions were obtained by choosing $\log(m_r/H(t_0)) = 2$. When studying the evolution of $\xi$ we varied the initial string density. Meanwhile, when analyzing the spectrum and energies we fixed the initial string density close to the scaling solution. With our method of generating initial conditions, this happens for $\xi(t_0) = 10^{-2}$ and $\xi(t_0) = 0.2$ for fat and physical strings respectively.

## A.2   String Length and Boost Factors

To identify strings and calculate their length, we adopt the algorithm proposed in Appendix A.2 of [22]. This involves counting the plaquettes that are pierced by a string, and converting the result to a length using a statistical correction factor. In doing so it is assumed that the strings are equally distributed in all directions.[21]

We calculate the boost factor $\gamma$ in two ways, the first as in Appendix A.2 of ref. [22] and the second as in ref. [31]. Briefly, the first method estimates the string velocity from the relativistic contraction of the string core, extracted from the derivative of the field on the gridpoints near the center of the string. The second instead measures the speed at which the points $\vec{x}$ such that $\phi(t, \vec{x}) = 0$ change in time. Both methods give the (local) $\gamma$-factor at each gridpoint where a string is identified. The frequency distribution function $\frac{d\xi_\gamma}{d\gamma}$ of $\gamma$-factors throughout the string network, defined in Appendix B.2, can be calculated easily. The average $\gamma$-factor of the network is defined as the mean over all the gridpoints where a string is identified. We checked that both methods give approximately consistent results, however the method of

---

[21]The results for the network match those of our previous algorithm (Appendix A.2 of [7]), up to a $\sim 5\%$ overall difference. We attribute this difference both to a small overcounting of our old method, and to a possible small violation of the isotropy due to the discrete grid. On the other hand, the methods give different results for individual string loops that are aligned in one particular direction, or when the density of strings is so small that the assumption of isotropy fails.

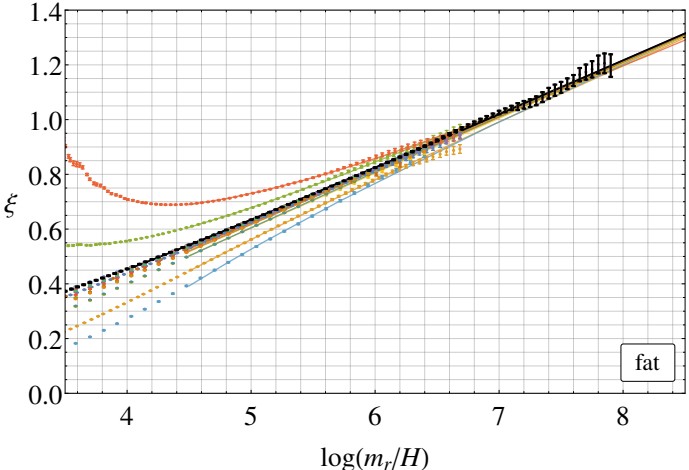

Figure 6: The evolution of the density of the fat string network, $\xi$, starting from different initial conditions, with statistical error bars. The convergence towards a common attractor solution, and the logarithmic growth of $\xi$ on this, are manifest. The best fit lines with the form eq. (3) are also shown. We identify the initial conditions used for the analysis of the spectrum of axions in black.

ref. [31] leads to superluminal $\gamma$-factors on some grid points, which have to be discarded in the counting.[22] Therefore we base our analysis on results using the method of ref. [22].

## B   Properties of the Scaling Solution and Log Violations

In this Appendix we discuss the properties of the scaling solution in more detail. We emphasize how the logarithmic violations of the naive scaling law affect different observables, such as the number of strings per Hubble volume, the relativistic boost factor of the network, the energy emitted in heavy radial modes and the axion spectrum. The dependence on $\log(m_r/H)$ of these properties (along with the supporting evidence from the dynamics of single loops studied in Appendix C) points to a consistent picture where the heavy degrees of freedom slowly decouple from the string network in the limit $\log(m_r/H) \to \infty$.

### B.1   The Scaling Parameter

The evolution of scaling parameter $\xi$ provides one of the clearest pieces of evidence of the attractor solution, and of the logarithmic violations of its scaling properties. Both of these features are already manifest for the physical string system in Fig. 1 and are even more evident in the fat string one.

In Fig. 6 we show $\xi$ for the fat string system as a function of $\log(m_r/H)$ with different initial string densities. For each initial condition we ran multiple simulations to reduce statistical errors. Thanks to larger cosmic time available to reach the same value of log, the results for $\xi$ converge to the attractor at smaller values of the log. The growth of $\xi$ on the attractor solution appears linear over a substantial range of log.

As discussed in the main text, the time-dependence of the scaling parameter is fit well by a universal linear function plus corrections proportional to powers of $1/\log$. The latter

---

[22]This is a drawback of the way the second method works: for instance, if a shrinking elliptic loop is very eccentric, its vertices are mistakenly interpreted as traveling at a very high speed when it is about to vanish. In the limit of infinite eccentricity, they would travel at infinite velocity.

encode the residual dependence on the initial conditions, and vanish in the large log limit. Including the first two such corrections, we perform a global fit of all of the $\xi$ data (separately for physical and fat strings) with the function in eq. (3), where $c_0$ and $c_1$ are universal while $c_{-1,-2}$ are let differ for each set of initial conditions. We include only points with log > 4.5 and log > 4 for fat and physical strings respectively and weight with the statistical errors.[23] For the latter we rescaled the $\chi^2$ function so as to have one independent contribution for every Hubble $e$-folding; this is in order to avoid bias from data set with a finer time sampling and decorrelate data from consecutive time shots. The result of the fit is fairly good, with a reduced $\chi^2_{\text{phys}} \approx 1.1$ and $\chi^2_{\text{fat}} \approx 1.5$. This indicates that eq. (3) is sufficient to capture the evolution of $\xi$ for the entire broad range of initial conditions considered. By the end of the simulations the fitted values of the parameters are such that the $1/\log$ corrections are already subleading.[24] This is particularly true for the fat string network simulations that converge to the attractor solution at smaller values of log. As it is clearly noticeable in Fig. 6, for the initial conditions closest to the attractor solution (these correspond to the black data set, which has the smallest values of $c_{-1,-2}$), $1/\log$ corrections are already negligible for the entire range of log plotted. Indeed, any fitting functions with sizable nonlinearities at late times is highly disfavored.

The coefficient of the linear log term $c_1$ is particularly important for the extrapolation to the physically relevent regime. The results for this in the fat and physical string systems are[25]

$$c_1^{\text{phys}} = 0.24(2), \qquad c_1^{\text{fat}} = 0.20(2). \tag{9}$$

Finally, we note that it has previously been shown that the percentage of the total string length in loops with size smaller than Hubble stays constant in time [7]. This means that the logarithmic violation in $\xi$ are reflected in a corresponding increase in the string length contained in small loops as well as long strings. This provides another strong piece of evidence that logarithmic violations are a genuine feature of the scaling solution.

All the simulations in Figures 1 and 6 have $0.5 \leq m_r \Delta \leq 1$, and the curves that reach later times are the average of multiple sets of simulations with different values of $m_r \Delta$. For any given initial condition the results from simulations with different $m_r \Delta$ give compatible results. This suggests that $\xi$ is already close to the continuum limit for $m_r \Delta = 1$ at least up to log $\sim 8$. Similarly, since the higher resolution simulations are stopped at $HL = 1.5$, when the simulations with poorer resolution have $HL \gg 1$, the agreement indicates that $\xi$ is close to the infinite volume limit for $HL = 1.5$.

## B.2 String Velocities

Another important quantity characterizing the dynamics of the network is the boost factors of the strings. Indeed, if strings are relativistic with an average boost $\langle \gamma \rangle$, their energy per unit length is increased by a factor $\langle \gamma \rangle$.[26] The theoretical expectation for the string tension $\mu \simeq \pi f_a^2 \log(m_r/H)$, which holds for strings at rest, is correspondingly modified to $\mu = \langle \gamma \rangle \pi f_a^2 \log(m_r/H)$. Therefore, the value and possible log dependence of $\langle \gamma \rangle$ must be understood so that the energy densities during the scaling regime can be determined correctly.

---

[23]Since the most overdense set in the physical case reaches scaling only late, we include data from this set at log > 5.5 in the global fit.

[24]If data at smaller logs is included higher corrections are needed to get a good fit. Meanwhile, selecting only data at larger logs still leads to a good fit but with greater uncertainties on the coefficients.

[25]These are compatible with those reported in our previous analysis [7], in which we studied the network up to log = 6.7.

[26]We always refer to the transverse boost, which does not lead to relativistic contraction of the string length. Consequently our definition of $\langle \gamma \rangle$, which does not have any extra weighting, gives the appropriate modification to the string tension.

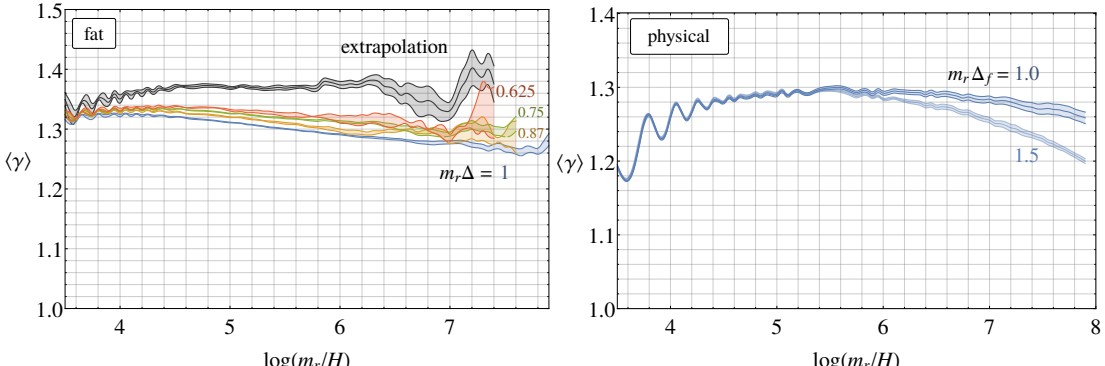

Figure 7: The dependence of the mean string boost factor, $\langle\gamma\rangle$, on $\log(m_r/H)$ for different lattice spacings, for fat (left) and physical strings (right). In the fat string case the continuum extrapolation is also plotted. In the physical case, the number of grid points per string core decreases with time, and the value of $m_r\Delta_f$ indicated is at the final time $\log(m_r/H) = 7.9$. For both the fat and physical string systems $\langle\gamma\rangle = 1.3 \div 1.4$ throughout, suggesting that the string network remains on average only mildly relativistic as the scale separation increases.

In Figure 7 we show the time evolution of the network's average $\gamma$-factor for fat and physical strings and for different lattice spacings (computed as described in Appendix A.2). Evidently the boost factor is lattice spacing dependent, with $\langle\gamma\rangle$ smaller for coarser lattices. This is not surprising given that the boost factor is measured by the size of the string cores, which might not be resolved when they are relativistically contracted. The continuum extrapolation indicates that, at least for fat strings, $\langle\gamma\rangle$ seems to asymptotically approach a constant mildly relativistic value $\langle\gamma\rangle = 1.3 \div 1.4$.[27]

More detailed information about string velocities can be inferred from the $\gamma$-factor distribution function. We define $\xi_\gamma$ to be the portion of $\xi$ with boost factor smaller than $\gamma$. Then $\frac{1}{\xi}\frac{d\xi_\gamma}{d\gamma}$ describes the distribution function of $\gamma$-factors in the network; $\langle\gamma\rangle = \int d\gamma\,\gamma\,\frac{1}{\xi}\frac{d\xi_\gamma}{d\gamma}$ being its first moment.

In Figure 8 we plot the velocity distribution function at different times for fat strings (already extrapolated to the continuum limit) and physical strings (for two different lattice spacings). The distribution is strongly peaked at nonrelativistic boost factors at all times with a sharp fall off $\propto \gamma^{-6}$. This makes the lowest moments of the distribution dominated by low boosts factors and explains why $\langle\gamma\rangle$ appears time independent. On the other hand, the large boost tail of the distribution keeps extending to increasingly large values at later times. Meanwhile, systematic errors from the finite lattice spacing affect the distribution at large $\gamma$ more at late times.[28]

This evolution with the log can be understood as follows: $\langle\gamma\rangle$ is dominated by long strings, which are mostly nonrelativistic (due to causality and Hubble friction) and make up the majority of the length at all times (see Section 3.3 of [7]). Therefore the network remains on average only mildly relativistic. Loops with size much smaller than Hubble provide a small, constant, proportion of $\xi$, but they get more relativistic as the log increases. These contribute to the high-$\gamma$ tail of the distribution, so this extends to larger $\gamma$ when the log is bigger. Similarly, kinks and cusps from string recombination also contribute. Such a logarithmic scaling

---

[27]The continuum limit plotted has been carried out with a linear extrapolation to zero lattice spacing. A quadratic extrapolation gives compatible results.

[28]This can be seen from Figure 8 (right): for $\log = 7$ the high $\gamma$-tail actually lies below that of $\log = 5.5$. However, the increasing spread between the $m_r\Delta = 1$ and $m_r\Delta = 1.5$ simulations indicates that this is a lattice effect, and the extrapolated tail at $\log = 7$ would be above that at $\log = 5.5$.

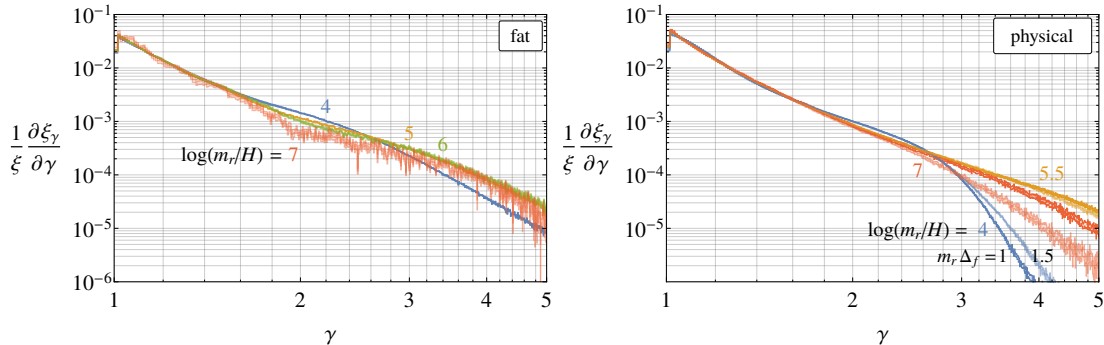

Figure 8: Left: The $\gamma$-factor distribution function $\frac{1}{\xi}\frac{d\xi_\gamma}{d\gamma}$ for fat strings (already extrapolated to the continuum limit) at different times. Right: The same function for physical strings for two different lattice spacings ($m_r \Delta_f = 1$ has the darker color, $m_r \Delta_f = 1$ the lighter). At all times, the distribution is peaked at $\gamma = 1$ with a sharp fall off $\propto \gamma^{-6}$ above this, so that $\langle\gamma\rangle$ is nonrelativistic. As the log increases, the high-$\gamma$ tail grows, suggesting that sub-horizon loops become increasingly relativistic.

violation is in agreement with the results in Appendix C where we show that single loops with larger initial logs get boosted more as they shrink.

We assume that the behavior identified above persists at large values of the log, and in particular that the mean $\gamma$-factor remains approximately constant. In this case the theoretical prediction for the string tension $\mu \simeq \pi f_a^2 \log(m_r/H)$ can be extrapolated to the physical scale separation (up to an overall $\langle\gamma\rangle \approx 1.3 \div 1.4$ constant factor, which we fit at small logs in the next Subsection).

### B.3 Effective String Tension and Radial Mode Decoupling

We now show that the string tension calculated in the simulation is in agreement with the theoretical expectation. Moreover, we show that the percentage of energy in radial modes, although non-negligible for small logs, decreases at late times, signaling the decoupling of heavy modes in the limit log → ∞.

The total energy density of the complex scalar field $\rho_{\text{tot}} \equiv \langle T_{00} \rangle$, where $T_{00}$ is the Hamiltonian density from the Lagrangian in eq. (1), can be split into components as

$$\rho_{\text{tot}} = \rho_s + \rho_a + \rho_r \,, \tag{10}$$

where $\rho_s$ is the energy density in strings, $\rho_a$ that in axion radiation and $\rho_r$ that in radial modes. $\rho_a$ is extracted from the kinetic energy density of the axion field $2\langle\frac{1}{2}\dot{a}^2\rangle$ away from string cores, and $\rho_r$ from the energy density of the radial field $\langle\frac{1}{2}\dot{r}^2 + \frac{1}{2}|\nabla r|^2 + V(r)\rangle$, again away from string cores (see [7] for more details). We then obtain $\rho_s$ from the difference $\rho_s = \rho_{\text{tot}} - \rho_a - \rho_r$. The string energy $\rho_s$ calculated in this way is expected to match the one predicted from the theoretical expectation for the string tension and the measured values of $\xi(t)$ up to order one coefficients.

We compute the effective tension $\mu_{\text{eff}} = \rho_s(t)t^2/\xi(t)$ from the definition in eq. (2), using $\xi(t)$ and $\rho_s(t)$ from simulations. This is then compared to the theoretically expected form

$$\mu_{\text{th}} = \langle\gamma\rangle \pi f_a^2 \log\left(\frac{m_r\,\eta}{H\sqrt{\xi}}\right), \tag{11}$$

accounting for a non-zero $\gamma$-factor, and for the dependence of the average inter-string distance on the string density (via the factor $1/\sqrt{\xi}$ in the log). The coefficient $\eta$ encodes the string

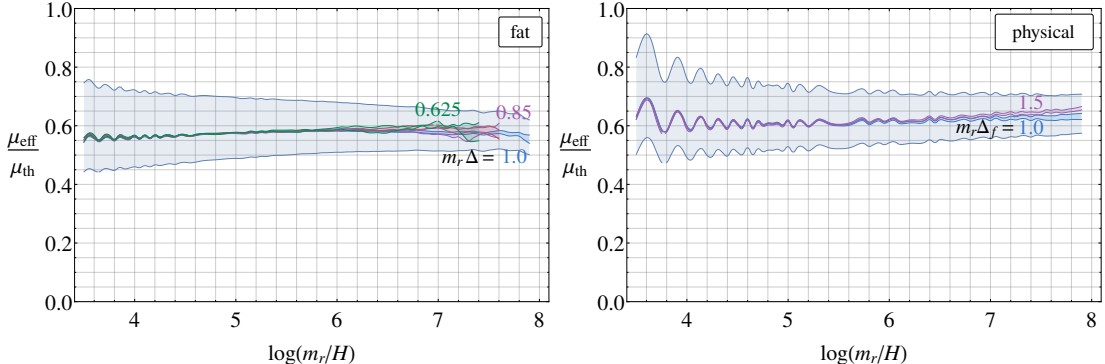

Figure 9: The ratio between the string tension calculated in simulations and the theoretical expectation in eq. (11) for fat (left) and physical strings (right). In both cases results are shown for different lattice spacings, and the blue shaded band indicates the effect of varying the parameter $\eta$ in eq. (11) over the range $[\frac{1}{2}, 2](1/\sqrt{4\pi})$. The approximately constant value of $\mu_{\text{eff}}/\mu_{\text{th}}$ for the whole simulation time suggests that our method of extracting the string energy density is consistent.

shape, and we chose $\eta = 1/\sqrt{4\pi}$ as a reference (we pick this somewhat arbitrarily based on the average distance between strings if they were all parallel, but any roughly similar value would also be reasonable).

In Figure 9, we plot the ratio $\mu_{\text{eff}}/\mu_{\text{th}}$ as a function of time for the fat string and the physical systems, where $\langle\gamma\rangle$ in $\mu_{\text{th}}$ is calculated in simulations as in the previous Section. Different colors represent different lattice spacings, and the blue shaded region shows the effect of varying $\eta$ in the interval $[\frac{1}{2}, 2](1/\sqrt{4\pi})$. For both fat and physical strings the tension measured in the simulation and the theoretical expectation are close over the whole time range. The $30 \div 40\%$ difference is not unexpected, given that strictly speaking eq. (11) only applies for straight strings and we do not have a reliable way to compute $\eta$ analytically. Instead, its value is determined by the loop distribution and the shape of the strings. Although $\langle\gamma\rangle$ and $\rho_s$ are rather sensitive to lattice spacing effects, $\mu_{\text{eff}}/\mu_{\text{th}}$ involves the ratio of the two and seems to have smaller systematic error. The approximately constant behaviour in Figure 9 gives us confidence that eq. (11) can be used to calculate the string tension at large logs.

In Figure 10 we plot the proportion of the total energy that is in radial modes as a function of time for different lattice spacings, i.e. $\rho_r/\rho_{\text{tot}}$. We also show the continuum extrapolation in the fat case.[29] The results for fat strings reveal an important feature: As the log increases the fraction of the total energy in radial modes decreases. Lattice spacing effects become increasingly significant, so this behaviour is only seen after the continuum extrapolation. Systematic errors from the lattice spacing also have a significant (and possibly even greater) effect for simulations of physical strings. Even though such simulations have better resolution prior to the final time, lattice spacing effects create a fake increase from $\log = 6$, which is shallower for the data with better resolution. Meanwhile, we will see in Appendix B.4 that the slight difference between the initial values of $\rho_r/\rho_{\text{tot}}$ for the two resolutions for the physical strings is due to a small difference in the initial conditions.

To sustain the scaling regime, energy density in strings is continuously emitted into axions and radial modes. We denote the emission rates, i.e. the energy released per unit time, by $\Gamma_a$ and $\Gamma_r$ respectively. These can be be computed from

$$\Gamma_i = \frac{1}{R^{z_i}} \frac{\partial (R^{z_i} \rho_i)}{\partial t}, \qquad (12)$$

---

[29]The continuum extrapolation shown is carried out with a linear extrapolation to zero lattice spacing, a quadratic extrapolation gives compatible results.

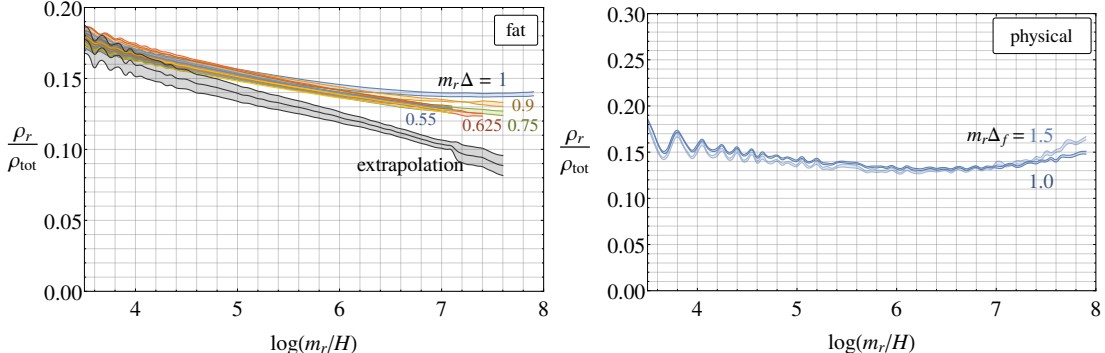

Figure 10: The fraction of total energy density in simulations that is in radial modes for physical strings (right) and fat strings (left). The results are shown for different lattice spacings, and the continuum extrapolation is also plotted for the fat string system. As log grows this fraction decreases for fat strings (after the continuum extrapolation), which suggests that radial modes decouple and are increasing irrelevant for the string dynamics. The results for physical strings show a similar trend, although we do not perform the continuum extrapolation.

where $i = a, r$, and $z_a = z_r = 4$ for fat strings due to the dependence of the radial mode mass on time.

In Figure 11 we plot the proportion of the total energy that is instantaneously emitted from strings that goes into axions, i.e. $r_a \equiv \Gamma_a / (\Gamma_a + \Gamma_r)$, for the fat string system. Similarly to the energy in radial modes, the rate of emission into axions is increasingly sensitive to the lattice resolution as the log grows (with $r_a$ larger for finer lattices). The continuum extrapolation suggests that $r_a$ increases steadily.

Together, the log dependences identified above indicate that the radial mode plays a decreasing role in the dynamics at late times in simulations. They also suggest that it should decouple in the limit $\log \to \infty$. Further, since only axions will get excited in this limit, the details of the particular UV physics that gives rise to the axion field would be unimportant for the dynamics of the strings.

## B.4 The Spectrum

As discussed in Section 2, the axion energy density spectrum, and its dependence on log, plays a key role in determining the axion number density when its mass becomes cosmologically relevant.

To define the axion spectrum we start from the expression for the axion energy density $\rho_a = \langle \dot{a}^2 \rangle$,

$$\rho_a = \frac{1}{L^3} \int d^3 x_p \, \dot{a}^2(x_p) = \frac{1}{L^3} \int \frac{d^3 k}{(2\pi)^3} |\tilde{a}(k)|^2 \,, \tag{13}$$

where $x_p = R(t)x$ are physical coordinates, and $\tilde{a}(k)$ is the Fourier transform of $\dot{a}(x_p)$. The axion spectrum $\partial \rho_a / \partial |k|$ is then fixed by requiring $\int d|k| \, \partial \rho_a / \partial |k| = \rho_a$, and is therefore given by

$$\frac{\partial \rho_a}{\partial k} \equiv \frac{\partial \rho_a}{\partial |k|} = \frac{|k|^2}{(2\pi L)^3} \int d\Omega_k |\tilde{a}(k)|^2 \,, \tag{14}$$

where $\Omega_k$ is the solid angle. In order to exclude strings, the field $a(x)$ needs to be screened. We substitute $\dot{a}(x) \to \dot{a}_{scr}(x) \equiv \left[1 + \frac{r(x)}{f_a}\right] \dot{a}(x)$ in eq. (13), since the factor $1 + \frac{r(x)}{f_a} = \frac{|\phi|}{|\phi|_{r \to 0}}$

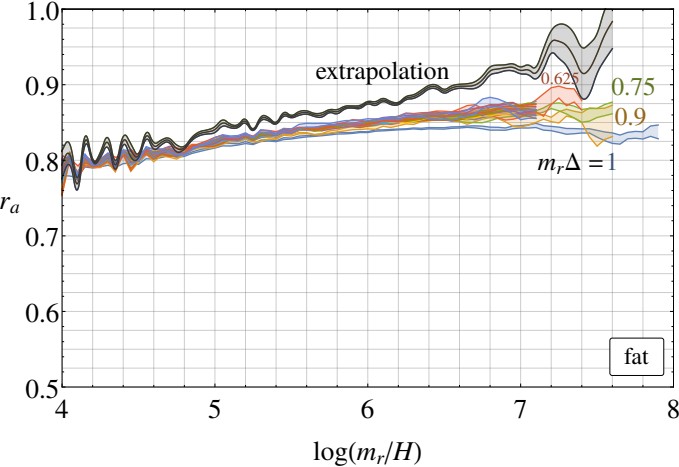

Figure 11: The ratio $r_a$ between the instantaneous energy emission rate from strings to axions and the total emission rate (i.e. to both axions and radial modes), as a function of log for different lattice spacings for fat strings. The continuum extrapolation shows that the network increasingly emits energy to axions (rather than to radial modes) as the log increases. This is consistent with the expectation that heavy modes decouple in the large log limit.

automatically vanishes inside string cores and tends to unity far from strings.[30]

In Figure 12 we plot the axion spectrum $\partial \rho_a / \partial k$ at different times for fat and physical strings. The initial conditions are close to the scaling solution, and correspond to the black lines in Figures 1 and 6. In both cases, the spectrum has a peak at momenta of order $5H \div 10H$ and at smaller $k$ it is power-law suppressed $\propto k^3$. Moreover, the spectrum has a UV-cutoff at momenta $k = m_r/2$, above which it is highly suppressed. The shape of the spectrum remains similar as time passes, modulo the shift in the UV-cutoff.

At the momentum $k = m_r/2$ there is a small peak, which we attribute to the energy exchange between axions and radial modes via parametric resonance. Such an effect can be understood heuristically. From eq. (8) in flat spacetime, the axion equation of motion is $(1 + \sigma) \partial_\mu \partial^\mu \theta + 2 \partial_\mu \sigma \partial^\mu \theta = 0$, where $\theta \equiv a/f_a$ and $\sigma \equiv r/(f_a/\sqrt{2})$.[31] In the presence of a spatially homogeneous radial mode $\sigma = \sigma_0 \sin(m_r t)$ with $\sigma_0 < 1$, the axion Fourier modes therefore satisfy $\ddot{\theta}_k^2 + k^2 \theta_k + 2\sigma_0 m_r \cos(m_r t) \dot{\theta}_k = 0$, where we kept the first non-vanishing dependence on $\sigma_0$. If $\sigma_0 \neq 0$, this is a parametric resonance equation for the mode $k = m_r/2$. A similar effect also occurs for a non-homogeneous radial field, as it is in simulations. For fat strings, $m_r$ decreases proportionally to the scale factor and the parametric resonance affects a unique comoving momentum at all times, as seen in Figure 12 (left). On the other hand, for physical strings a wide range of comoving momenta are affected, and the resulting oscillations cover almost the entire spectrum. As discussed in Appendix A, this effect is easily triggered if the strings in the initial conditions do not have a core-size equal to $m_r^{-1}$. In this case, radial modes will be emitted while the string core size is adjusting, and these will produce axions.[32] As the string cores relax the rate of such emission will decrease, and the parametric resonance effect will gradually disappear. This can be seen from the reduction in the amplitude of the oscillations in the final time shots plotted in Figure 12.

We also compute the energy density spectrum of radial modes $\partial \rho_r / \partial k$. Since radial modes behave as massive waves, this can be defined similarly to the axion spectrum as

---

[30]As checked in [7], this method reproduces the Pseudo Power Spectrum Estimator introduced in [85] well.

[31]See also eq. (42) with $m_a = 0$.

[32]Of course in this case the radial mode will be space-dependent and the discussion above does not strictly apply.

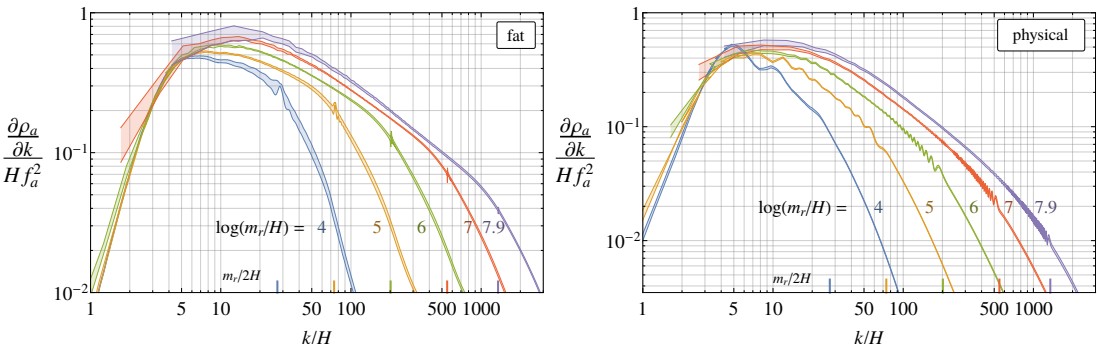

Figure 12: The axion energy density spectrum at different times (i.e. at different value of $\log(m_r/H)$) for the fat (left) and physical (right) string systems, as a function of the momentum $k$ in units of Hubble. In both cases the spectrum is dominated by a broad peak at around $k/H = 10$, and emission at lower momenta is suppressed. For each time shot we also show the value of $k = m_r/2$, corresponding to the parametric resonance frequency with the radial mode. There is little energy in modes with momenta corresponding to scales smaller than the string cores. The physical simulations have $m_r \Delta = 1$ at the final time, and the fat simulations have $m_r \Delta = 1$ throughout.

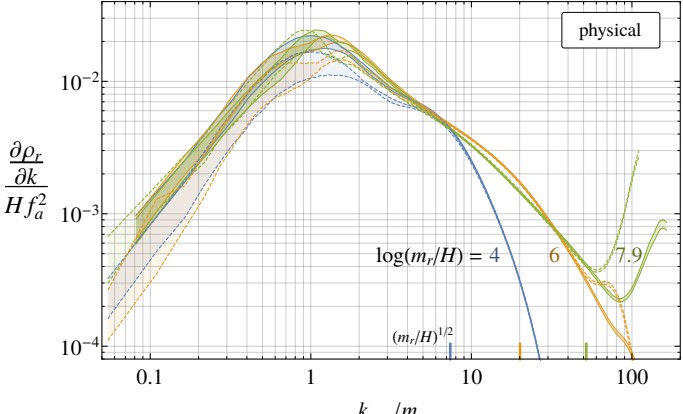

Figure 13: The energy density spectrum of radial modes for the physical string system as a function of the comoving momentum $k_{\text{com}} \equiv k\sqrt{m_r/H}$ at different times. Solid lines represent results from simulations with $m_r \Delta_f = 1$, and dashed lines with $m_r \Delta_f = 1.5$. The spectrum is dominated by an IR peak that comes from the initial conditions, with modes at larger comoving momenta generated during the evolution. We also indicate the comoving momentum that corresponds to $m_r$ at each of the times plotted.

$\int dk \partial \rho_r / \partial k \equiv \langle \dot{r}^2 \rangle$, so

$$\frac{\partial \rho_r}{\partial k} = \frac{|k|^2}{(2\pi L)^3} \int d\Omega_k |\tilde{\dot{r}}(k)|^2 \,. \tag{15}$$

To avoid strings in the determination of the radial spectrum we adopt the same masking technique as for the axion, i.e. in eq. (15) we substitute $\dot{r}(x) \rightarrow \dot{r}_{\mathrm{scr}}(x) \equiv \left[1 + \frac{r(x)}{f_a}\right] \dot{r}(x)$.

In Figure 13 we plot the spectrum of radial modes $\partial \rho_r / \partial k$ for physical strings at three different times and for two lattice spacings, $m_r \Delta_f = 1$ and 1.5. The spectrum is plotted as a function of the comoving momentum $k_{\mathrm{com}} \equiv k\sqrt{m_r/H}$ and we also indicate the momentum corresponding to the radial mode mass at each time.

Figure 13 reveals interesting features of the evolution of radial modes. First, the spectrum is peaked at a fixed comoving momentum, corresponding to $k = m_r$ at the time $H = m_r$, and it has a sharp fall off at momenta bigger than $m_r$ at any given time. The average momentum is therefore smaller than $m_r$ at all times and radial modes are on average nonrelativistic. Second, the height of the peak decreases proportionally to $H \propto R^{-2}$, i.e. as in the free nonrelativistic limit. This shows that this peak is entirely produced at early times and does not receive contributions afterwards. Therefore the slight differences in $\rho_r/\rho_{\mathrm{tot}}$ at low momenta observed in Figure 10 for the two lattice spacings are only due to the initial conditions. Finally, the lattice spacing effects at high momenta grow at larger logs, producing a fake rise in simulations with the worse resolution. This is in turn related to the fake growth of the ratio $\rho_r/\rho_{\mathrm{tot}}$ in Figure 10 at late times.

### B.4.1 The Instantaneous Emission Spectrum

We now turn to study the spectrum with which axions are instantaneously emitted by the network. To do so, we extract $F$ from its definition $F(k/H) = \partial \log(\Gamma)/\partial(k/H)$ of Section 2.2 and express $\Gamma$ in terms of the axion spectrum as in eq. (12). This leads to

$$F\left[\frac{k}{H}, \frac{m_r}{H}\right] = \frac{A}{R^3} \frac{\partial}{\partial t} \left(R^3 \frac{\partial \rho_a}{\partial k}\right) \,. \tag{16}$$

In the last equation $A = H/\Gamma$, which is a consequence of the normalization condition $\int_0^\infty F[x, y] \, dx = 1$, which follows from its definition (see ref. [7] for more details). The time derivative in eq. (16) is calculated from simulation data by taking the difference of spectra with $\Delta \log = 0.25$.[33]

In Figure 14 we plot $F$ at different times. We see that $F[x, y]$ has IR and UV cutoffs at $x = 5 \div 10$ and $x = y/2$ for all $y$ (i.e. at all times). In between it is well approximated with a power law $1/x^q$. Moreover, similarly to the total spectrum, $F$ has significant fluctuations at the momenta affected by the previously described resonance, which for physical strings encompasses a large range of $x$ (as discussed these do not represent genuine emission from strings, but are an unphysical effect that will disappear in the large log limit).[34] Finally, in Figure 14 a change in the power-law $q$ with the log can be seen by eye. This corresponds to the evolution of $q$ plotted in Figure 2 of the main text. The momentum range $[30H, m_r/4]$ over which $q$ is calculated in Figure 2 is highlighted in Figure 14. The increase in $q$ is even clearer in Figure 15 where we plot $xF$. At the final simulation time the instantaneous emission is not far from reaching $q = 1$.

We now analyze the possible functional form of $q(\log)$ of Figure 2. The result of the linear and the quadratic fits are shown in Figure 2 in dark and light orange respectively. The two

---

[33]With this choice the statistical fluctuations are still relatively small, while the value of $F$ is already compatible with the one of the $\Delta \log \rightarrow 0$ limit.

[34]If initial conditions that are not sufficiently close to scaling are used, the fluctuations dominate the instantaneous emission for the physical system to such an extent that it will not show a clear power-law behaviour.

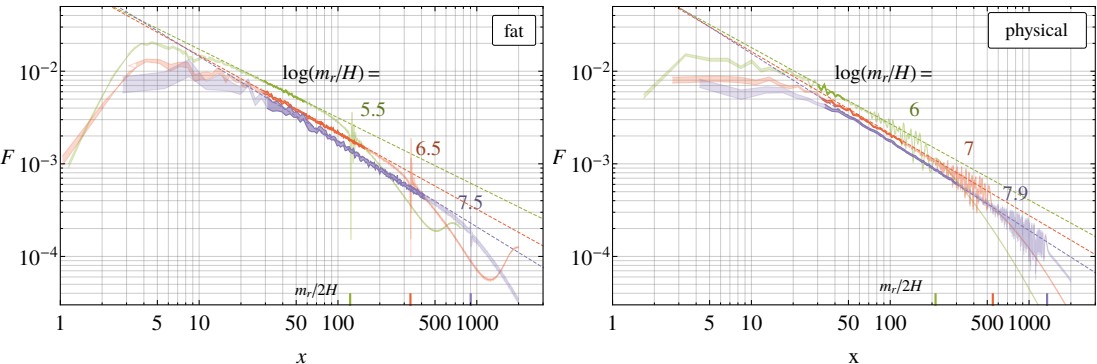

Figure 14: The instantaneously emitted axion energy density spectrum $F[x, y]$ as a function of $x = k/H$ at different times (represented by $y = m_r/H$) for fat strings (left) and physical strings (right). We also plot dotted lines corresponding to the best fit values of $q$ for each time (obtained by fitting the $q$ as a linear function of log over the complete data set, leading to eq. (17)). An increase in the slope $q$ with log can be seen for both the fat and physical string systems. The highlighted region corresponds to the data points in the range $[30H, m_r/4]$, which we use for the fit of $q$ in Section 2. We also indicate $m_r/(2H)$ at each time, above which emission is highly suppressed.

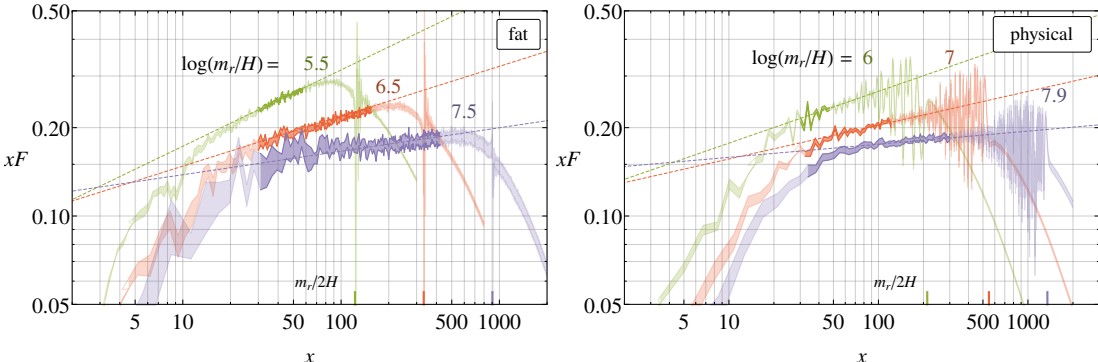

Figure 15: The instantaneously emitted axion energy density spectrum multiplied by $x$, i.e. $xF[x, y]$, as a function of $x = k/H$ and at different times for fat strings (left) and physical strings (right). The increase in $q$ is evident and at the final time the instantaneous emission is almost scale invariant ($q = 1$). It is reasonable to expect that at $\log \simeq 8$ the slope $q$ will overtake the value $q = 1$.

fits are both quite good for the fat and also the physical case. However, the result from the quadratic fit is compatible with the linear fit, but with larger errors. This suggests that a linear fit is enough to reproduce the data. For the linear fit $q = q_1 \log + q_2$, we get

$$
\begin{cases} q_{1\,\mathrm{phys}} = 0.053(5) \\ q_{2\,\mathrm{phys}} = 0.51(7) \end{cases} , \qquad \begin{cases} q_{1\,\mathrm{fat}} = 0.084(2) \\ q_{2\,\mathrm{fat}} = 0.28(2) \end{cases} . \tag{17}
$$

In both the physical and the fat string systems the fit return values of $q$ larger than unity for $\log \gtrsim 9$, which might be accessible with future generation simulations. In particular, for physical strings, the fit gives $q_{\mathrm{phys}}(\log \to 70) = 4.1(5)$.

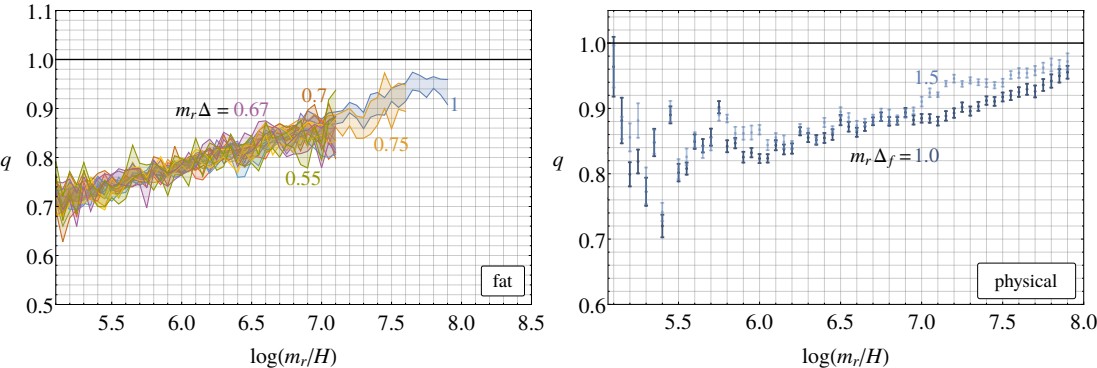

Figure 16: The best fit power-law $q$ as a function of log, for simulations with different lattice spacings, in the fat (left) and physical (right) string systems. The error bars shown are statistical. The results from different lattice spacings are compatible in the fat string case and show a clear increase in $q$. For the physical case there are fluctuations due to energy transfer from radial modes at early times (discussed in the main text), but subsequently data from both resolutions shows a clear increase with the log.

### B.4.2 Systematics

Given the importance of $q$ for the final axion abundance, we now analyze potential systematic errors in its determination in detail.

**Lattice spacing effects.** For the fat case, we performed multiple sets of simulations with different resolutions (from $m_r \Delta = 1/1.8$ to $m_r \Delta = 1$), and in Figure 16 (left) we plot $q$ as a function of log for each set. We calculate $q$ by fitting the slope of $\log F$ (as a function of $\log x$) in each simulation over the momentum range $[30H, m_r/4]$, and then averaging over simulations with the same resolution. The result for $q$ is compatible for all lattice spacings. Consequently, in Figure 2 in the main text we report the average of all the sets. The largest values of log, i.e. $7.7 < \log < 7.9$, can be explored only with the least conservative lattice spacing ($m_r \Delta = 1$), and therefore for those we have no direct comparison. However, given the good agreement for smaller logs, we expect that these values are in the continuum limit as well.

In the physical case we performed two sets of simulations, with final values of $m_r \Delta_f = 1$ and 1.5 (at earlier times the resolution is better). The result for $q$ (calculated as before) is shown in Figure 16 (right). Due to the parametric resonance effect, $q$ has unphysical fluctuations at small log. However, for $\log > 6$ the fluctuations are relatively minor and a growth in $q$ is clear for both resolutions. Moreover, the results for the two lattice spacings are mostly compatible with each other. Nevertheless, given the slight difference, in the main text we reported only the results with $m_r \Delta_f = 1$, which is the more conservative.

**UV and IR cutoffs.** In extracting $q$, the extremes $k_{IR}$ and $k_{UV}$ of the momentum range fitted $[k_{IR}, k_{UV}]$ need to be sufficiently far from the Hubble peak and the UV cutoff respectively. By construction, $q$ is therefore less prone to lattice spacing and finite volume systematics compared to other quantities, such as energy densities (which are more UV sensitive) and number densities (which are more IR sensitive). For the fat string system, in Figure 17 we plot $q$ at different times as $k_{IR}$ and $k_{UV}$ are varied. This shows that $q$ has already converged to its true value with the choice $[30H, m_r/4]$.

In the physical case the best fit value of $q$ has a small dependence on the momentum range

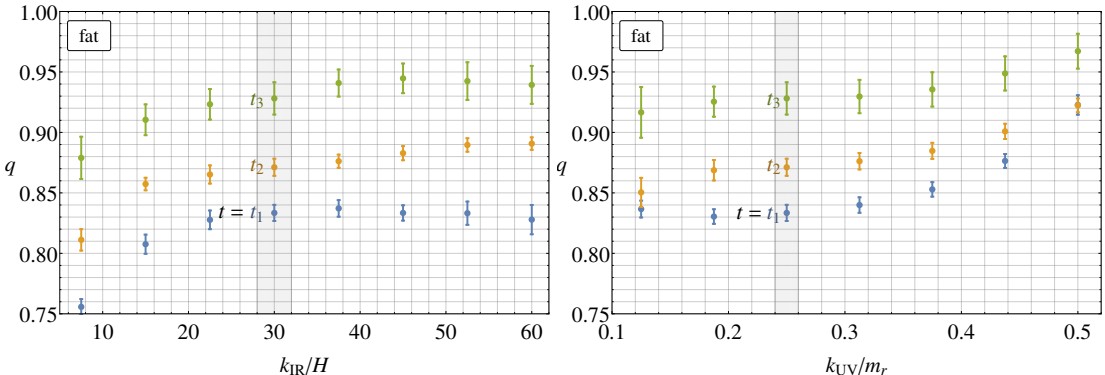

Figure 17: Left: The best fit values of $q$ for fat strings at three different times (different colors) for varying $k_{IR}$, with $k_{UV} = m_r/4$ fixed. Right: The best fit $q$ with varying $k_{UV}$ and $k_{IR} = 30H$ fixed. It can be seen that $k_{IR} = 30H$ and $k_{UV} = m_r/4$ are sufficient for $q$ to have convergered to its true value. This analysis has been done for $m_r\Delta = 1$ and the times $t_1, t_2, t_3$ correspond to $\log = 6.5, 7, 7.5$.

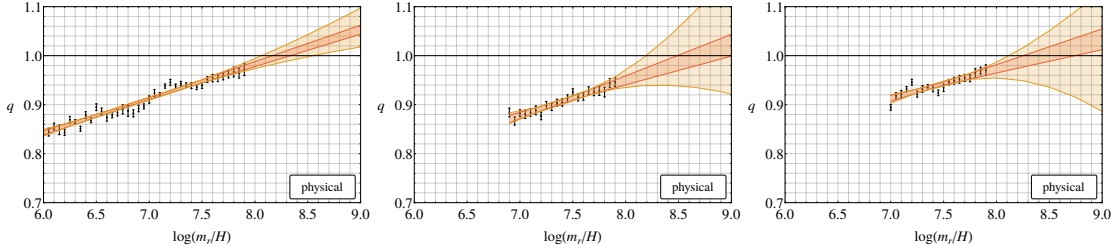

Figure 18: The best fit values of $q$ as a function of log for physical strings, with statistical error bars. Left: results with $q$ fit over the momentum range $[30H, m_r/4]$ for data with lattice spacing at the final time $m_r\Delta_f = 1.5$. Center: $q$ fit in the range $[50H, m_r/6]$ for $m_r\Delta_a = 1$. Right: $q$ fit in the range $[50H, m_r/6]$ for $m_r\Delta_f = 1.5$. In all plots the light and dark red bands represent the best linear and quadratic fits to $q$ vs log with standard errors on the fit.

used. Figure 2 (left) in the main text shows the fit in $[30H, m_r/4]$ for $m_r\Delta = 1$. In Figure 18, we show different fits choosing $k_{IR}$ between $30H$ and $50H$ and $k_{UV}$ between $m_r/6$ and $m_r/4$, for the two different lattice spacings $m_r\Delta_f = 1$ and $1.5$. While the numerical value of $q$ at a particular log changes slightly, the choice of the momentum range does not change the trend of $q$ increasing with log. Indeed, in the same plots we also show the linear fit of $q$ as a function of log, and all of these have a positive gradient. We also show the quadratic fits, which are compatible with the linear fits but with much larger uncertainties.

**Finite Volume.** As mentioned in Appendix A, the simulations have been run until $HL = 1.5$. In Figure 19 we show $F$ for different choices of $HL$ at $\log = 6.4$ for the fat string system. While for $HL = 1.5$ the IR part of the spectrum gets modified compared to $HL = 2$, in the momentum range where $q$ is extracted the two are completely compatible. Although this is shown only for $\log = 6.4$, finite volume effects are not expected to depend strongly on log (and are also expected to be similar for the physical string system). Thus, the choice $HL = 1.5$ will not introduce a significant systematic error in the calculation of $q$.

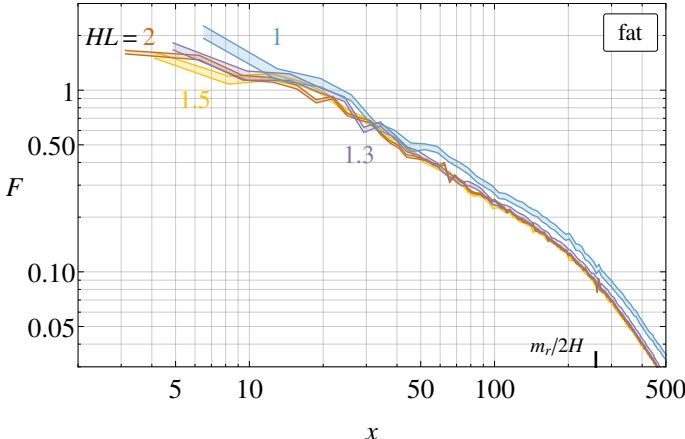

Figure 19: The instantaneous emission $F$ for different values of $HL$ at $\log = 6.4$. For $HL \geq 1.5$ finite volume effects do not alter the spectrum in the momentum range $[30H, m_r/4]$ where $q$ is extracted. The simulations are performed with $m_r \Delta = 1$ and a $N_x^3 = 1250^3$ grid.

## C  Log Violations in Single Loop Dynamics

In this Appendix we give additional evidence of logarithmic violations in the dynamics of global string by studying the collapse of circular loops in flat spacetime. Sub-horizon loops make up a small percentage of the total string length during the scaling solution [7]. However, we will see that they can still give useful insights into the properties of the scaling regime, especially in relation to the Nambu–Goto limit.

The dynamics of global string loops can be described by an effective theory in which the fundamental degrees of freedom are the string and the axion radiation [21], with an interaction governed by the Kalb–Ramond action [86]. This theory is valid in the regime $\log(m_r R) \gg 1$, where $R$ is the typical loop size, i.e. when the string and the emitted radiation (with frequency $\omega \sim 1/R$) are not strongly coupled. In particular, this action does not capture the dynamics when strings intersect.

As shown in [21], in this theory the coupling of the axion to the string is proportional to $1/\log(m_r R)$. So, in the limit $\log(m_r R) \to \infty$, the axion radiation decouples from the string. The string loop would therefore behave as in the free (Nambu–Goto) limit, oscillating an infinite number of times. As described in [36], this suggests that for finite but still large $\log(m_r R)$ the loop might bounce many times before disappearing, emitting radiation with a typical wavelength of the order of the initial loop size. It has been argued in [36] that this will lead to a spectrum with $q > 1$. However, the previous argument is not definitive because the effective theory breaks down when the loop has shrunk to a small size, and the dynamics when the loop is small are critical in determining whether it bounces.

A complete analysis of the evolution of a string loop, including the bounce, can be carried out by solving the full field equations with the heavy radial mode present. We numerically solved eq. (8) in Minkowski spacetime with initial conditions $\phi(x)$ and $\dot{\phi}(x)$ that resemble a static circular loop with initial radius $R_0$. Limitations on the gridsize require $\log(m_r R_0) \lesssim 5$. In Figure 20 we plot the loop radius $R(t)$ (normalized to its initial value $R_0$) as a function of time for different $\log(m_r R_0)$. We also show the free Nambu–Goto time law, $R_{NG}(t) = R_0 \cos(t/R_0)$.

Figure 20 has a number of interesting features. First, as $\log(m_r R_0)$ increases, $R(t)$ gets closer to the prediction for free strings.[35] As a result, the relativistic boost factor increases with

---

[35]Indeed as shown in Appendix F of [7], the EFT calculation of [21] reproduces the solution of the field equations

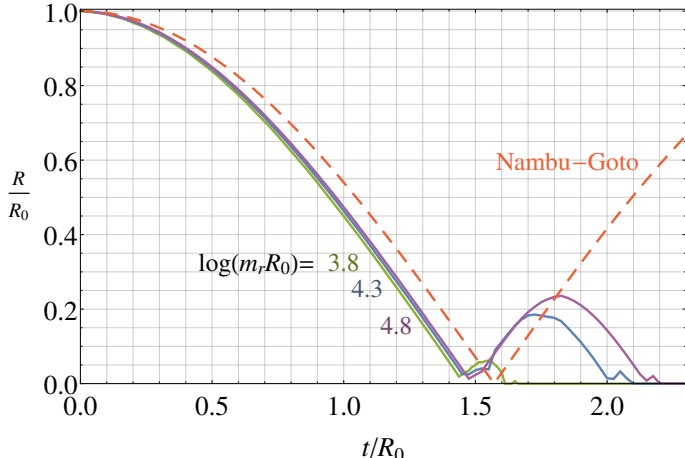

Figure 20: The radius $R(t)$ vs time for circular loops, normalized to the initial radius $R_0$. The different lines correspond to different values of the initial $\log(m_r R_0)$. The dashed red line is the free Nambu–Goto solution $R_{NG}(t) = R_0 \cos(t/R_0)$. As $\log(m_r R_0)$ grows, the time-law $R(t)$ approaches the Nambu–Goto limit, and the loop tends to bounce more. This is in agreement with the picture proposed in [36].

$\log(m_r R_0)$, and tends to infinity in the limit $\log(m_r R_0) \to \infty$. By taking the time derivative of $R(t)$ one can easily show that, for instance, the boost factor is already of order 10 for $\log(m_r R_0) = 5$. Second, the loop tends to bounce more for increasing $\log(m_r R_0)$: for example, a loop with $\log(m_r R_0) = 5$ oscillates producing a loop with $\log(m_r R_0) \approx 4$, which subsequently bounces to a loop with $\log(m_r R_0) \approx 2$. The larger bounce is likely to be related to the increased boost factor, because a relativistic string loop is less likely to release all its energy at once before disappearing.[36]

We also note that correctly evolving strings with a large boost factor requires a fine lattice spacing to resolve the relativistically contracted core and the bounce. For instance, the simulations need to be performed with $m_r \Delta = 1/20$ or smaller for $\log(m_r R_0) = 5$, otherwise the loop will unphysically collapse as soon as it approaches its center.[37]

After the loop disappears, the energy will be released into axions and heavy radial modes. As $\log(m_r R_0)$ increases, we checked that the percentage of the energy emitted in axions gets larger, again pointing to the decoupling of the radial mode in the large $\log(m_r R_0)$ limit.[38] Although not definitive because they are done at small logs, these results support the picture proposed in ref. [36], and agree with the general discussion of the previous Appendix.

# D  The End of the Scaling Regime

The scaling regime of Section 2 holds in the limit of vanishing axion potential, and it ends once the axion mass becomes cosmologically relevant, which happens when $H$ and $m_a$ are of the same order. In this Section we make the previous approximate expectation more precise, and show that the scaling regime is not affected by $m_a \neq 0$ until $H > m_a$. Indeed we identify

---

for $\log(m_r R_0) = 5$ well, at least when the loop has not yet collapsed and the EFT is under control.

[36]Indeed, the loop passes through itself during the oscillation, as we checked by calculating the sign of the phase of $\phi$ before and after the bounce.

[37]This unphysical decay was interpreted in [87] as a sign of string loops not approaching the Nambu–Goto limit. This decay is also related to a non-conservation of the total energy during the bounce.

[38]This has also been observed in single loop lattice simulations in [88].

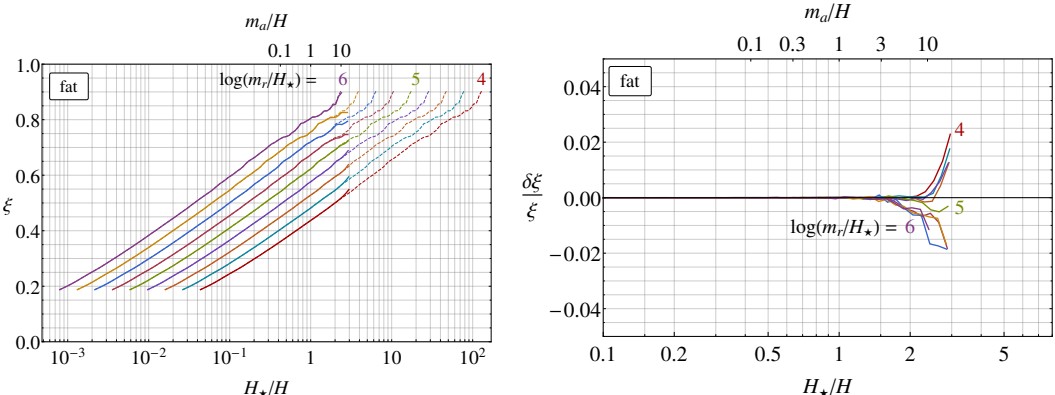

Figure 21: Left: The scaling parameter $\xi$ as a function of time when the axion mass is turned on at different $\log_\star \equiv \log(m_r^\star/m_a^\star)$ (solid lines), and for $m_a = 0$ throughout with the same initial condition (dashed lines). The x-axis is normalized with $H_\star$, so it can be seen that regardless of $\log_\star$, $\xi$ is unaffected by the mass before $H = H_\star$. Right: The relative difference between the results for $\xi$ with and without a non-zero axion mass, $\delta\xi/\xi$, as a function of time. We define $\delta\xi \equiv (\xi - \xi_{m_a=0})$, and $\xi_{m_a=0}$ is the string length in a simulation with $m_a = 0$ throughout. For all values of $\log_\star$ tested the effect of a non-vanishing axion mass is smaller than percent for $H > H_\star$.

$H = m_a$ as the point at which the scaling regime begins to break down.

The full Lagrangian of the system is the same as in Section 2 with the addition of a potential for the axion, which we take of the form

$$\mathcal{L} = |\partial_\mu \phi|^2 - V(\phi), \qquad \text{with} \quad V(\phi) = \frac{m_r^2}{2f_a^2}\left(|\phi|^2 - \frac{f_a^2}{2}\right)^2 + m_a^2 f_a^2\left(1 - \frac{\text{Re}[\phi]}{f_a/\sqrt{2}}\right), \quad (18)$$

where $\phi = \frac{r+f_a}{\sqrt{2}}e^{i\frac{a}{f_a}}$ and $m_a^2$ is the time dependent axion mass introduced in eq. (4). The equation of motion from eq. (18)

$$\ddot{\phi} + 3H\dot{\phi} - \frac{\tilde{\nabla}^2 \phi}{R^2} + \phi \frac{m_r^2}{f_a^2}\left(|\phi|^2 - \frac{f_a^2}{2}\right) - \frac{m_a^2 f_a}{\sqrt{2}} = 0, \quad (19)$$

does not depend on $f_a$ directly. Instead, the dependence on the two scales $f_a$ and $m_r$ can be reabsorbed by rescaling the field $\phi \to \phi f_a$ and the space-time coordinates $t \to t/m_r$ and $x \to x/m_r$. Therefore, up to a trivial field rescaling, the physics is only sensitive to the two ratios $m_r/H = 2m_r t$ and $m_r/m_a$, which we will refer to in the following. We implement eq. (19) numerically in the fat string system, described in Appendix A.

As in Section 3, we define $H_\star$ to be the Hubble parameter at the time when $H = m_a$, and correspondingly $m_a = H_\star(H_\star/H)^{\alpha/4}$ and $\log_\star \equiv \log(m_r^\star/H_\star)$, where $m_r^\star \equiv m_r(t_\star)$. The choice of $\log_\star = \log(m_r^\star/m_a^\star)$ determines the scale separation at $H = H_\star$, sets the time at which the axion mass becomes relevant, and fixes the axion mass in units of $m_r$ in eq. (19). Although $\log_\star \approx 60 \div 70$ for physical axion masses, we can study the breaking of the scaling regime only for $\log_\star \lesssim 6$. To do so we analyze when the evolution of the system with finite $\log_\star$ starts deviating from the evolution with the same initial conditions but with $m_a = 0$ throughout.

In Figure 21 (left) we show the evolution of the scaling parameter $\xi$ with time for axion masses with $\alpha = 7$ and different choices of $\log_\star$ (solid curves). We also show the evolution for vanishing axion mass (dashed curves). The initial conditions are the same in all of the simulations and are on the scaling solution. Figure 21 (right) shows the relative difference between the curves with and without the axion mass as a function of time. The choice of $H_\star/H$

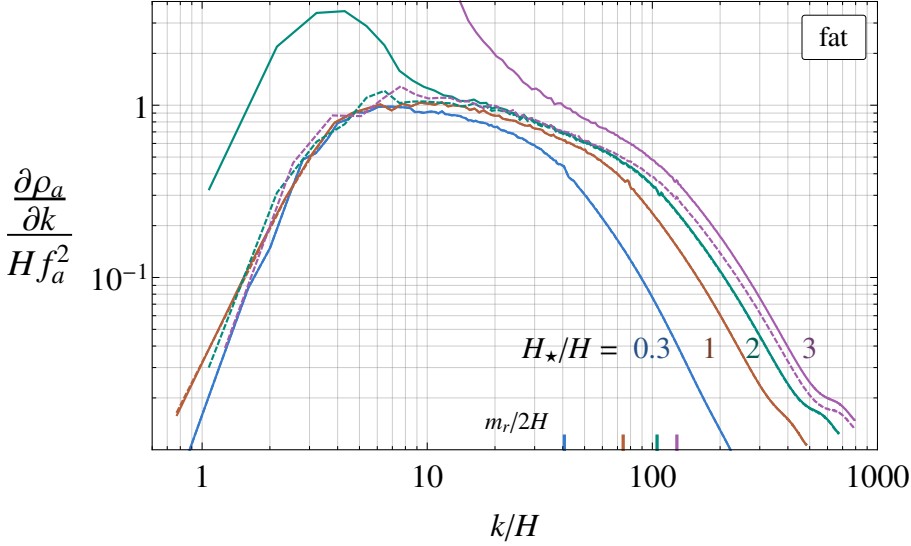

Figure 22: The evolution of the axion spectrum in the presence of the axion mass with $\log_\star = 5$ and $\alpha = 6$ (upper lines) at different times labeled by $H_\star/H$, and for vanishing axion mass (lower lines, dashed). Before $H = H_\star$ the non-zero axion mass has a negligible effect on the spectrum, but subsequently this gets modified starting starting from IR momenta.

on the $x$-axis makes it clear that the critical Hubble at which $\xi$ starts differing by more than one percent is $H_{\text{crit}} = H_\star$ for all $\log_\star$. We stopped evolving these simulations at $m_a/H \simeq 10$, since at later times systematic errors due to an insufficient hierarchy between $m_a$ and $m_r$ can become significant (discussed in Appendix F). We have checked however that at later times the effect of a nonzero axion mass is to reduce the string length, until the whole network disappears.

In Figure 22 we show the axion energy spectra at different times (i.e. different $H_\star/H$) for a system with an axion mass such that $\log_\star = 5$ with $\alpha = 7$. We again plot the results obtained from a system with $m_a = 0$ (dashed lines) for comparison. While $H > H_\star$ there is no significant difference between the two spectra, and at $H < H_\star$ the spectrum starts differing substantially. We note that a nonzero axion mass affects the spectrum earlier than it affects $\xi$. In particular the IR modes, which contribute the most to the axion number density, are affected first and change soon after $H = H_\star$. At later times UV modes are also affected. The results obtained are similar for other values of $\log_\star$.

We have checked that the behavior seen in Figures 21 and 22 holds for all values $0 \leq \alpha \leq 8$. As expected, for larger $\alpha$ the mass affects $\xi$ and the spectrum even less while $H > H_\star$, but the network shrinks within fewer Hubble times after this point.

We therefore conclude that before the critical value $H_{\text{crit}} = H_\star$ the string system is not affected by a non-vanishing axion mass. The accuracy of our results is not sufficient to establish whether $H_{\text{crit}}$ itself has a small residual dependence on $\log_\star$, and indeed this is plausible considering the log violations discussed in Appendix B. Possible violations are expected to be small since the axion mass changes rapidly, and would therefore not significantly affect the lower bound on the axion abundance calculated in Section 3.

As discussed in Section 4, we do not attempt to calculate the axion number density that is emitted by the string and domain wall network as the latter is destroyed after $m_a = H$. Indeed, we do not expect that a direct calculation in simulations would reproduce the number density in the physically relevant regime. In particular, at the values of log accessible in simulations

the strings are still emitting a UV dominated spectrum and have a tension and density $\xi$ that is far from the physically relevant values.

# E   Axions through the Nonlinear Regime

In this Appendix we will give further details on the derivation of the analytic prediction for the axion number density after its potential becomes relevant. We also describe the simulations that we performed to confirm its validity and fit its free parameters.

## E.1   Derivation of the Analytic Estimate

The initial conditions for the axion equations of motion of the Hamiltonian in eq. (4),

$$\ddot{a} + 3H\dot{a} - \frac{\tilde{\nabla}^2 a}{R^2} + m_a^2 f_a \sin\left(\frac{a}{f_a}\right) = 0, \qquad m_a = H_\star \left(\frac{H_\star}{H}\right)^{\alpha/4}, \tag{20}$$

correspond to a superposition of waves with energy spectrum $\frac{\partial \rho_a}{\partial k}$ described in detail in Appendix B.4, emitted by strings during the scaling regime prior to $H = H_\star$. As discussed below, the initial field can be obtained inverting eq. (16) as a function of $\xi$ and $F$ at $H = H_\star$. In doing so we assume that at large log the energy from strings is emitted purely into axions (as the results in Appendix B.3 suggest). The extrapolation of the effective string tension $\mu_{\text{eff}}$ is also needed. As shown in Appendix B.3, $\mu_{\text{eff}}$ in simulations is reproduced well by the theoretical expectation $\mu_{\text{th}} = \langle \gamma \rangle \pi f_a^2 \log\left(\frac{m_r}{H} \frac{\eta_c}{\sqrt{\xi}}\right)$ with $\eta_c = 1/\sqrt{4\pi}$ and $\gamma \approx 1.3$ constant in time, and we assume that this remains approximately true also at large scale separations.

The actual form of $F$, shown in Fig. 14, has a nontrivial shape. For simplicity here we approximate it with a single power law $q$ and a sharp IR cutoff at $x < x_0$,

$$F[x, y] = \begin{cases} \frac{(q-1)x_0^{q-1}}{x^q} & x \in [x_0, y] \\ 0 & x \notin [x_0, y] \end{cases}. \tag{21}$$

As we will see, the results obtained from this simple form capture the main features of the dynamics. We also considered more complex shapes reproducing the one in Fig. 14 more closely. However, when compared to numerical simulations, the improvement with respect to the simplified approximation in eq. (21) is negligible once compared to the uncertainties induced by the large $\log_\star$ extrapolation.

By inverting eq. (16), the total energy in axions will be distributed with the spectrum

$$\frac{\partial \rho_a}{\partial k}(t, k) = \int^t dt' \frac{\Gamma'}{H'} \left(\frac{R'}{R}\right)^3 F\left[\frac{k'}{H'}, \frac{m_r}{H'}\right], \tag{22}$$

where the primed quantities are computed at the time $t'$, the redshifted momentum is defined as $k' = kR/R'$ (see eq. (23) of [7] for the explicit derivation). As mentioned in Section 2.2, the emission rate $\Gamma \simeq \xi \mu_{\text{eff}}/t^3 \simeq 8\pi H^3 f_a^2 \xi \log(m_r/H)$ is fixed by energy conservation, and we assume a linear logarithmic growth of $\xi$, as in eq. (3), and $q > 1$. Using eq. (21), the resulting

convoluted spectrum at $H = H_\star$ for momenta $k < x_0 \sqrt{H_\star m_r}$ is[39]

$$
\begin{aligned}
\frac{\partial \rho_a}{\partial k}(t_\star, k) =\ & \frac{8 \xi_\star \mu_\star H_\star^2}{k} \left[ \left(1 - 2 \frac{\log(k/k_0)}{\log_\star}\right)^2 - \left(\frac{k_0}{k}\right)^{q-1} \right. \\
& \left. + 4 \frac{1 - 2 \frac{\log(k/k_0)}{\log_\star} - \left(\frac{k_0}{k}\right)^{q-1}}{(q-1)\log_\star} + 8 \frac{1 - \left(\frac{k_0}{k}\right)^{q-1}}{(q-1)^2 \log_\star^2} \right],
\end{aligned}
\tag{23}
$$

where $k_0 = x_0 H_\star$. For $k > x_0 \sqrt{H_\star m_r}$ the spectrum falls faster than $1/k$ and its precise form is not important since it gives a negligible contribution to the abundance. To a good approximation, we can neglect the effect of $\gamma$, $\eta_c$ and the order one factor between $\mu_{\text{eff}}$ and $\mu_{\text{th}}$ of Figure 9, so we take $\mu_\star \approx \pi f_a^2 \log_\star$.[40] The terms in the last line of eq. (23) are $\mathcal{O}(1/\log_\star)$ and can be neglected in the large $\log_\star$ limit, so in this limit all the dependence of $\left.\frac{\partial \rho_a}{\partial k}\right|_\star$ on $q$ is also suppressed.

Due to the scaling regime, the leading dependence of the spectrum for $k > x_0 H_\star$ is $\left.\frac{\partial \rho_a}{\partial k}\right|_\star \propto 1/k$ for all $q \geq 1$ (i.e. the spectrum obtained after convoluting $F$ is scale invariant). Correspondingly, the energy is distributed equally in logarithmic intervals between the momenta $x_0 H_\star$ and $\sqrt{H_\star m_r}$. The logarithmic dependence of $\xi_\star$ and $\mu_\star$ on time induces violations of the scale invariance that are proportional to $\log^2(k/k_0)$.

At least up until $H = H_\star$, away from strings axions propagate as free waves, and their spectrum can therefore be used to infer the axion field itself via the relations (valid for relativistic waves)

$$
\rho_a \equiv \int dk \frac{\partial \rho_a}{\partial k} = \frac{1}{V} \int \frac{d^3 k}{(2\pi)^3} \left| \dot{\tilde{a}}(\vec{k}) \right|^2 = \frac{1}{V} \int \frac{d^3 k}{(2\pi)^3} k^2 \left| \tilde{a}(\vec{k}) \right|^2 ,
\tag{24}
$$

where $k = |\vec{k}|$ and $\tilde{a}(\vec{k})$ is the Fourier transform of $a(t_\star, \vec{x})$ and $V$ is the volume. From the last equality of eq. (24) it also follows that the average square amplitude is

$$
\langle a^2 \rangle \equiv \frac{1}{V} \int_V d^3 x \, a^2(x) = \frac{1}{V} \int \frac{d^3 k}{(2\pi)^3} \left| \tilde{a}(\vec{k}) \right|^2 = \int \frac{dk}{k^2} \frac{\partial \rho_a}{\partial k} .
\tag{25}
$$

Combining this with eq. (23) we obtain $\langle a^2 \rangle|_\star = 4 \xi_\star \mu_\star$ for $\log_\star \gg 1$ and $\xi_\star \log_\star \gg 1$. Consequently, as mentioned, $\langle a^2 \rangle|_\star$ is much larger than $f_a^2$ in this limit.

Following the procedure sketched in Section 3, we can now derive the formula for the contribution to the final abundance from the spectrum of eq. (23). In particular, in the limit of large $\log_\star$, the effects from the axion potential can be neglected[41] also after $t_\star$ and the spectrum evolves relativistically as

$$
\frac{\partial \rho_a}{\partial k}(t, k) = \frac{8 \xi_\star \mu_\star H^2}{k} \left[ \left(1 - 2 \frac{\log\left[\frac{k H_\star^{1/2}}{k_0 H^{1/2}}\right]}{\log_\star}\right)^2 - \left(\frac{k_0 H^{1/2}}{k H_\star^{1/2}}\right)^{q-1} \right] .
\tag{26}
$$

This evolution holds up until the contribution from the axion potential in the Hamiltonian ($\rho_V = m_a^2(t) f_a^2$) becomes of the same order as the gradient one from IR nonrelativistic modes,

---

[39]In principle we should include a dependence of $q$ on log to calculate the spectrum ($q \propto \log y$ in eq. (21)). However, as we will see below, provided $q \gg 1$ its precise value and its dependence on time are not important.

[40]More precisely, if the extrapolation of $\mu_{\text{eff}}/\mu_{\text{th}}$ of Figure 9 remains valid at large log, not taking these effects into account induces a $\sim 20\%$ overestimation of the energy density.

[41]This is reminiscent of the so-called kinetic misalignment introduced in [89, 90].

i.e. until $t = t_\ell$ when the following condition is satisfied:

$$\rho_{\text{IR}}(t_\ell) \equiv \int_0^{c_m m_a(t_\ell)} dk \frac{\partial \rho_a}{\partial k} = c_V m_a^2(t_\ell) f_a^2 \,, \tag{27}$$

where $c_V$ and $c_m$ are $\mathcal{O}(1)$ coefficients to be determined numerically.

The condition above provides the following implicit equation for $t_\ell$, or equivalently for $m_a(t_\ell)/H_\star$:

$$8\xi_\star \mu_\star H^2 \left[ \log(\kappa)\left(1 - 2\frac{\log(\kappa)}{\log_\star} + \frac{4}{3}\frac{\log^2(\kappa)}{\log_\star^2}\right) - \frac{1-\kappa^{1-q}}{q-1}\right] = c_V m_a^2 f_a^2 \,, \quad \kappa = \frac{c_m m_a}{x_0 \sqrt{HH_\star}} \,, \tag{28}$$

where all the quantities are evaluated at $t = t_\ell$. We note in particular that $\rho_{\text{IR}}$ is still much larger than $m_a^2 f_a^2$ at $H = H_\star$ because of the enhancement by a factor of $\xi_\star \mu_\star \propto \log_\star^2$.

Introducing the quantity

$$z \equiv \left(\frac{m(t_\ell)}{H_\star}\right)^{1+\frac{6}{\alpha}} \,, \tag{29}$$

which measures the delay of the nonlinear regime induced by the $\xi_\star \log_\star$ enhancement, eq. (28) can be rewritten as

$$8\pi\xi_\star \log_\star \left[ \log(\kappa)\left(1 - 2\frac{\log(\kappa)}{\log_\star} + \frac{4}{3}\frac{\log^2(\kappa)}{\log_\star^2}\right) - \frac{1-\kappa^{1-q}}{q-1}\right] = c_V z^{2\left(1-\frac{2}{\alpha+6}\right)} \,, \quad \kappa = \frac{c_m}{x_0} z^{1-\frac{4}{\alpha+6}} \,. \tag{30}$$

The equation can be further simplified by noticing that the first term dominates in the limit $\log_\star \gg 1$, and thus we get

$$8\pi\xi_\star \log_\star \log\left(\frac{c_m}{x_0} z^{1-\frac{4}{\alpha+6}}\right) = c_V z^{2\left(1-\frac{2}{\alpha+6}\right)} \,. \tag{31}$$

Eq. (31) can be solved analytically using the identity $a\log(bz^c) = z \iff z = -acW_k(-(acb^{1/c})^{-1})$ for some $k \in \mathbb{Z}$, where $W_k(z)$ is the Lambert $W$-function evaluated on the $k$-th Riemann sheet and defined by $ze^z = a \iff z = W_k(a)$. The solution is

$$z = \left[\frac{W_{-1}\left(-\frac{c_V\left(1+\frac{2}{\alpha+2}\right)}{4\pi\xi_\star \log_\star}\left(\frac{x_0}{c_m}\right)^{2\left(1+\frac{2}{\alpha+2}\right)}\right)}{-\frac{c_V\left(1+\frac{2}{\alpha+2}\right)}{4\pi\xi_\star \log_\star}}\right]^{\frac{1}{2}\left(1+\frac{2}{\alpha+4}\right)} \,, \tag{32}$$

where the choice of the lower branch $k = -1$ is dictated by the fact that the argument of $W$ is negative (because $c_m, c_V > 0$) and the value of $W_k$ in eq. (32) must be negative (and large) so that $z > 1$. In the limit $\xi_\star \log_\star \gg 1$ we can expand $W_{-1}$ for small negative arguments. Noticing that

$$W(-z^{-1}) = \log\left(\frac{-z^{-1}}{W(-z^{-1})}\right) = \log\left(\frac{-z^{-1}}{\log\left(\frac{-z^{-1}}{\cdots}\right)}\right) = -\log(z\log(z\log(\cdots))) \,, \tag{33}$$

where the second equality is just the recursion of the first equality and the dots stand for infinitely nested logs, eq. (32) gives

$$z = \left[\frac{4\pi\xi_\star \log_\star}{c_V}\left[1 - \frac{2}{\alpha+4}\right]\log\left(\frac{4\pi\xi_\star \log_\star}{c_V}\left[1 - \frac{2}{\alpha+4}\right]\left[\frac{c_m}{x_0}\right]^{2\left(1+\frac{2}{\alpha+2}\right)}\log(\ldots)\right)\right]^{\frac{1}{2}\left(1+\frac{2}{\alpha+4}\right)} \,, \tag{34}$$

where the logarithms are infinitely nested. At $t = t_\ell$ the field dynamics is completely nonlinear with most of the spatial gradients of order the axion mass or lower and energy density of order $m_a^2(t_\ell)f_a^2$. As the Universe continues to expand, the energy density and the field value decrease further, and so do nonlinearities. We assume that the transient lasts $\mathcal{O}(1)$ Hubble times, so that during this period the total energy is approximately conserved. After the nonlinear transient, the axion field drops below $f_a$, the dynamics is mostly linear and the comoving number density is conserved again. The latter can therefore be derived from the energy density at $t_\ell$ as

$$n_a(t_\ell) = c_n \frac{\rho_{\mathrm{IR}}(t_\ell)}{m_a(t_\ell)} = c_n c_V m_a(t_\ell) f_a^2, \tag{35}$$

where the $\mathcal{O}(1)$ coefficient $c_n$, which we will fit from numerical simulations, parametrizes all matching effects during the nonlinear transient, such as the $\mathcal{O}(1)$ effects from the redshift during the transient and the extra contribution from slightly relativistic modes above $c_m m_a(t_\ell)$.

We finally arrive to an expression for the relative contribution of relic axions from strings during the scaling regime normalized to the reference misalignment value at $\theta_0 = 1$ (i.e. $n_a^{\mathrm{mis},\theta_0=1}(t_\ell) = c_n' m_a(t_\star)f_a^2(H_\ell/H_\star)^{3/2}$, where $c_n' \equiv 2.81$)

$$
\begin{aligned}
Q(t_\ell) &\equiv \frac{n_a^{\mathrm{str}}(t_\ell)}{n_a^{\mathrm{mis},\theta_0=1}(t_\ell)} \\
&= \frac{c_n}{c_n'} c_V \left[ \frac{W_{-1}\left( -\frac{c_V\left(1+\frac{2}{\alpha+2}\right)}{4\pi\xi_\star \log_\star} \left(\frac{x_0}{c_m}\right)^{2\left(1+\frac{2}{\alpha+2}\right)} \right)}{-\frac{c_V\left(1+\frac{2}{\alpha+2}\right)}{4\pi\xi_\star \log_\star}} \right]^{\frac{1}{2}\left(1+\frac{2}{\alpha+4}\right)} \\
&= \frac{c_n}{c_n'} c_V \left[ \frac{4\pi\xi_\star \log_\star}{c_V}\left[1 - \frac{2}{\alpha+4}\right] \log\left( \frac{4\pi\xi_\star \log_\star}{c_V}\left[1 - \frac{2}{\alpha+4}\right]\left[\frac{c_m}{x_0}\right]^{2\left(1+\frac{2}{\alpha+2}\right)} \log(\ldots) \right) \right]^{\frac{1}{2}\left(1+\frac{2}{\alpha+4}\right)}.
\end{aligned}
\tag{36}
$$

Conservation of the comoving number density implies that $Q(t)$ is constant for $t \gg t_\ell$, so that one can easily compute the number density of axions today, $n_a^{\mathrm{str}}(t_0)$, from the equation above.

From the analytic expression in eq. (36) we can notice several important features. First, as a result of the large $\log_\star$ approximation, the explicit dependence on $q$ has disappeared (we will show below from the full numerical results how indeed such a dependence is subleading). Second, the dependence on $x_0$ (the effective IR scale of the spectrum) is only logarithmic. This softens considerably systematic errors from neglecting a possible evolution of $x_0$ during the scaling regime, and makes manifest the insensitivity of the final result on the details of the shape of the IR part of the spectrum. Third, the unknown parameters $c_{V,m,n}$ enter the final formula as multiplicative factors (or in the logarithmic dependence), so that the functional dependence on $\xi_\star \log_\star$, $\alpha$ and $x_0$ is a true prediction of the above analytic derivation, which is confirmed by the numerical computations. Notice finally that with $\alpha \gg 1$, i.e. when the axion potential grows fast and the redshift effects between $t_\star$ and $t_\ell$ can be neglected, the formula aboves simplifies further, recovering the simple dependence $Q(t_\ell) \propto (\xi_\star \log_\star)^{1/2}$, anticipated in the main text. In Appendix E.3 we will see that eq. (36) fits well the numerical results and we will provide numerical fits for the coefficients $c_{V,m,n}$ (see the caption of Fig. 24), which indeed are of order one.

## E.2   Setup of the Numerical Simulation

In Section 3 we also studied the evolution of the axion field numerically by solving eq. (20) on a discrete lattice. Unlike our simulations of the scaling regime, eq. (20) only contains the axion

and not the radial mode. The absence of the radial mode means that strings are automatically absent, and simulations can reach values of $H_\star$ that are arbitrarily smaller than $f_a$ and $m_r$.

The numerical implementation of eq. (20) is very similar to the string system. In particular, it is convenient to use the rescaled field $\psi = R(t)a/f_a$ and the conformal time $\tau$, which is defined as

$$\tau(t) = \int_0^t \frac{dt'}{R(t')} \propto t^{1/2} \, . \tag{37}$$

In this way eq. (20) simplifies to

$$\psi'' - \tilde{\nabla}_x^2 \psi + u(\tau)\sin\left(\frac{\psi}{R}\right) = 0 \, , \qquad u(\tau) = \left(\frac{\tau}{\tau_\star}\right)^{\alpha+3} \, , \tag{38}$$

where $\psi'$ and $\tilde{\nabla}_x \psi$ are derivatives with respect to the dimensionless variables $H_\star\tau$ and the comoving coordinate $H_\star x$, and $\tau_\star = \tau(t_\star)$.

We solve eq. (38) numerically starting from $\tau = \tau_\star$ in a box with periodic boundary conditions. Space is discretized on a cubic lattice of comoving side length $L_c$ containing up to $N_x^3 = 3000^3$ uniformly distributed grid points. The space-step between grid points in comoving coordinates $\Delta_c = L_c/N_x$ is constant in time. Eq. (38) is discretized following a standard central-difference Leapfrog algorithm for wave-like partial differential equations, and evolved in fixed steps of conformal time $\Delta_\tau$, which we fix to $\Delta_\tau = \Delta_c/3$.[42] The derivatives are expanded to fourth order in the space-step and second order in the time-step.

The physical length of the box is $L(t) = L_c R(t)$ and the physical space-step between grid points $\Delta(t) = L/N_x = \Delta_c R(t)$ grows $\propto t^{1/2}$. Therefore, the number of Hubble patches in the box $HL$ decreases with time, and the axion mass in lattice units $m_a\Delta$ rapidly increases. This leads to potential systematic uncertainties in the results, which we analyze in Section E.4. Our final results are obtained with parameter choices that are free from significant uncertainties.

The initial conditions $a(t_\star, \vec{x})$ and $\dot{a}(t_\star, \vec{x})$ are extracted from $\left.\frac{\partial\rho_a}{\partial k}\right|_\star$ via their Fourier transforms from eq. (24). More precisely, eq. (24) fixes only $\langle|\tilde{a}(\vec{k})|^2\rangle$ (and the time derivative), i.e. the average over the points in a sphere in Fourier space with fixed radius $|\vec{k}|$. For a fixed wave-vector $\vec{k}$, we therefore generated $|\tilde{a}(\vec{k})|^2$ randomly in the interval $\langle|\tilde{a}(\vec{k})|^2\rangle\left[1 - \vec{k}^2/k_{max}^2, 1 + \vec{k}^2/k_{max}^2\right]$, where $k_{max} \equiv 2\pi N_x/L$ and $\langle|\tilde{a}(\vec{k})|^2\rangle$ is a function of $|\vec{k}|$ fixed from eq. (24). Similarly, the phase of $\tilde{a}(\vec{k})$ is chosen randomly in $[0, 2\pi)$ for all $\vec{k}$.

In our analysis we used two different functional forms for the initial spectrum $\left.\frac{\partial\rho_a}{\partial k}\right|_\star$. The first is that of eq. (23), which is derived from the simplified $F$ consisting of a single power law. For the final results in the main text, we used a more realistic spectrum obtained by inserting a better approximation to the true shape of $F$ (plotted in Figure 14). This comprises four different power laws, chosen to be $x^3, x^{1/2}, x^{-1/2}, x^{-q}$ joined at the points $x = x_0/4, x_0, 2x_0, y$, and with $F[x > y, y] = 0$. The intermediate power laws in this reproduce the broad IR peak in $F$. For both functional forms we used $q = 5$, as suggested by the extrapolation of $q(\log_\star)$ of eq. (17) at $\log_\star = 70$. As shown below, provided $q \gg 1$ its precise value is not important for large $\xi_\star\log_\star$, so this choice does not have any significant effect on the results obtained.[43]

## E.3  Further Results from Simulations and Oscillons

To understand the dynamics as the axion mass becomes cosmologically relevant it is useful to study the mean square amplitude $\langle a^2\rangle$ defined in eq. (25), the axion spectrum $\frac{\partial\rho_a}{\partial k}$ and the

---

[42]As shown in ref. [7] this time discretization is sufficient to avoid numerical effects at the per-mille level.

[43]The initial spectrum was always generated for $\log_\star = 70$, and different $\xi_\star\log_\star$ are obtained by varying $\xi_\star$ (so, e.g. the normalization of eq. (23) changes, but not the $k$-dependence). This choice has no significant effect for $\xi_\star\log_\star \gg 1$.

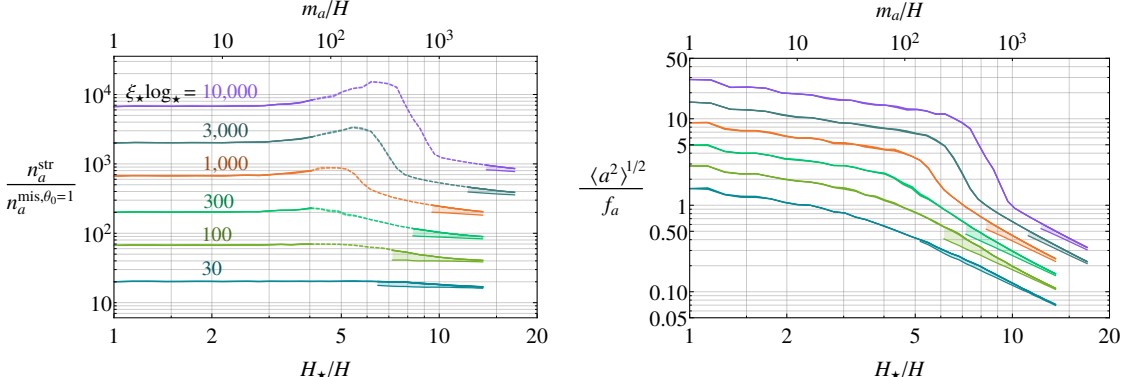

Figure 23: The time evolution of the comoving number density (left) and of the mean square axion field amplitude (right) for different values of $\xi_\star \log_\star$, with $x_0 = 10$ and $\alpha = 8$. The colors on the right figure correspond to the same values of $\xi_\star \log_\star$ as on the left. The lower lines, below the shaded regions, are the same observables computed with oscillons masked, which only makes sense at late times once these are well defined objects. It can be seen that after a transient the comoving axion number density approaches a constant value. Meanwhile, at early times the mean field amplitude evolves as relativistic matter, and at late times as nonrelativistic matter, as expected.

axion number density

$$\frac{\partial \rho_a}{\partial k} \equiv \frac{\partial \rho_a}{\partial |\vec{k}|} = \frac{|\vec{k}|^2}{(2\pi L)^3} \int d\Omega_k \left[ \frac{1}{2} |\tilde{\dot{a}}(\vec{k})|^2 + \frac{1}{2} \left(\vec{k}^2 + m_a^2\right) |\tilde{a}(\vec{k})|^2 \right],$$
$$n_a^{\mathrm{str}} = \int \frac{dk}{\sqrt{k^2 + m_a^2}} \frac{\partial \rho_a}{\partial k}. \tag{39}$$

In Figure 23 we show the time evolution of the mean square axion field amplitude $\langle a^2 \rangle$ and of the comoving number density $Q(t) = n_a^{\mathrm{str}}(t)/n_a^{\mathrm{mis},\theta_0=1}(t)$. The results are shown for different values of $\xi_\star \log_\star$, and with the simplified initial axion spectrum with IR cutoff $x_0 = 10$ and with $\alpha = 8$. As mentioned in Section 3, for sufficiently large $\xi_\star \log_\star$ the early evolution of these quantities matches the expectation of a relativistic regime. Calculating $n_a^{\mathrm{str}}$ makes sense at these times since the whole potential is negligible in the Hamiltonian density (4), which is diagonal in momentum space. After a transient period when nonlinearities dominate the evolution of the system and $n_a^{\mathrm{str}}$ is not defined (corresponding to the dashed lines in Figure 23), the average amplitude becomes smaller than $f_a$. At such times the majority of the field is in the linear regime, with the exception of objects called oscillons that are produced.

An oscillon is a localized, metastable, time-dependent solution eq. (20), in which the field oscillates with maximum amplitude of order $\pi f_a$ within a region of (decreasing) size $m_a^{-1}$ [46]. It therefore contains an energy density of order $\frac{1}{2} m_a^2 f_a^2 \pi^2$. Although oscillons decay, radiating their energy into axions, they are thought to be very long lived [91, 92]. Indeed, we observe that they do not disappear within the range of our simulations. As time passes, however, they occupy an increasingly negligible proportion of space. Consequently it makes sense to calculate the number density of axions by considering only the field far away from oscillons, where the linear approximation to eq. (39) is valid.

We screen oscillons by multiplying the axion field and its time derivative by a window function $w(\rho_a)$ of the local energy density $\rho_a(x)$ (defined as in eq. (4)), i.e. $a_{scr}(x) \equiv w(\rho_a(x)) a(x)$.

We choose the window function to be

$$w(\rho_a) = \frac{1}{2}\left[1 - \tanh\left[5\left(\frac{\rho_a}{\frac{1}{2}m_a^2 f_a^2 \pi^2} - 1\right)\right]\right], \tag{40}$$

which vanishes for $\rho_a \gtrsim \frac{1}{2}m_a^2 f_a^2 \pi^2$ and tends to 1 for $\rho_a \ll m_a^2 f_a^2$ as required. We have checked the results are not sensitive to this particular functional form. The screened average amplitude, spectrum and number density are defined as in eqs. (25) and (39) substituting $a(x) \to a_{scr}(x)$ and similarly for the time derivative.

In Figure 23 we plot the mean field amplitude and the axion number density at late times both with and without oscillon screening (respectively, the lower and upper curves between the shaded regions).[44] At earlier times (i.e. during the transient) we do not show the results with oscillons screened since they are not yet clearly defined objected. As expected, with oscillons screened the system reaches a regime in which the comoving number density is conserved, and the average field amplitude decreases as in the limit of nonrelativistic dynamics.[45] Moreover, the effect of oscillons decreases at late times. We have tested that the results at the end of simulations are not sensitive to the minimum energy density that is masked by the window function, provided this is in a reasonable range.

Despite their interesting dynamics, we refrain from analyzing the evolution of the oscillons in detail. Instead we just note a few relevant facts. We estimated the number of oscillons in two ways: (1) by dividing the total volume in oscillons (defined as the points where $w(\rho_a) > 1/2$) by the volume of one oscillons $m_a^{-3}$, and (2) by dividing the total energy in oscillons by the energy of one oscillon ($m_a^{-3} \times \frac{1}{2}m_a^2 f_a^2 \pi^2$). The number of oscillons computed in the two ways is compatible and constant in time. This means that, once formed, oscillons do not decay within the simulation time. Additionally, the number of oscillons that form depends on the initial amplitude of the field and increases if the value of $\xi_\star \log_\star$ in the initial conditions is larger.

Our analytic estimate for $Q$ has been derived only for the simplified spectrum in eq. (21). However, this differs from the more physical spectrum only in its IR part. As argued before, $Q$ is not very sensitive to this part of the spectrum, so that we can apply the result from eq. (36) to the more physical spectrum. We expect the different shape of the initial spectrum to be reabsorbed in the numerical fit of the coefficients $c_m, c_V, c_n$.

In Figure 24 and Figure 3 in the main text, we show the comoving number density of axions $Q(t_f)$ at the final simulation time $t_f$ as a function of $\xi_\star \log_\star$ for $\alpha = 4, 6, 8$ and $x_0 = 5, 10, 30$ for the physical initial spectrum. The errors are systematic and come from the uncertainty in the screening of oscillons, which we estimate as the difference between the masked and unmasked number density. The statistical errors, as well as the systematic errors from finite volume and finite UV-cutoff are subdominant. The continuous lines in the same plot represent the analytic estimate in eq. (36) (valid for $\xi_\star \log_\star \gg 1$), where the coefficients $c_m, c_V, c_n$ have been fixed with a global fit of all the data points with $\xi_\star \log_\star \geq 100$ in Figures 24 and 3. We note that the fit is good, and the dependence on $\xi_\star \log_\star$, $\alpha$ and $x_0$ in the numerical data is captured well by the analytic result. Equivalent plots for the simplified initial spectrum show a similarly good fit.

Finally, in Figure 25 we show the time evolution of the spectrum $\frac{\partial \rho_a}{\partial k}$ for $\xi_\star \log_\star = 10^3$ with $\alpha = 8$ and $x_0 = 10$. As expected, the spectrum evolves as in the free relativistic limit initially, and the nonlinear transient depletes the amplitude of modes $k < m_a(t_\ell)$. Oscillons affect the spectrum only at momenta of order of the axion mass at a given time, indicated with an empty dot.

---

[44]Strictly speaking, the number density in eq. (39) has no physical meaning if the field is not in the linear regime, i.e. when oscillons are not screened.

[45]Note that oscillons will continue radiating axions with momentum of order their inverse size, $k \sim m_a$. These will not contribute significantly to the final axion number density, as can be seen in the spectrum of Figure 4.

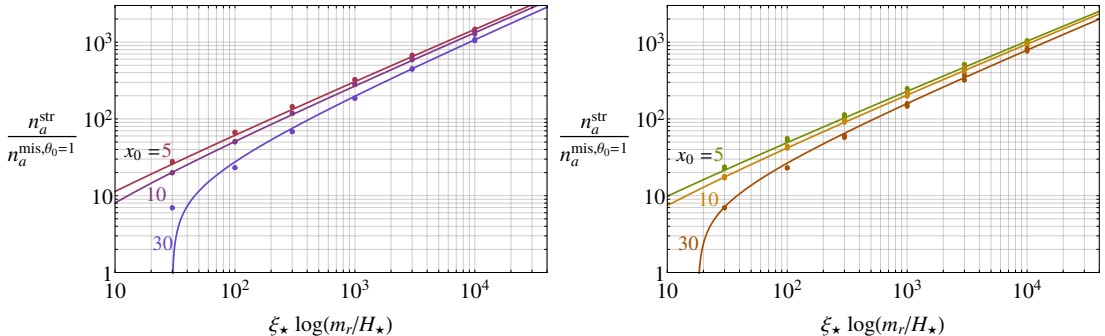

Figure 24: The ratio between the axion number density from strings and from misalignment (with $\theta_0 = 1$) as a function of $\xi_\star \log_\star$ for $\alpha = 4, 6$ (left and right panels respectively). The data points are simulation results, while the lines are the analytic estimate in eq. (36) with coefficients $c_m = 2.08$, $c_V = 0.13$, $c_n = 1.35$ fit from the complete data set. The analytic fit matches the data well, including the dependence on $x_0$ and $\alpha$ (except at very small $\xi_\star \log_\star$ where it is expected to break down).

For all of the results presented so far we have fixed $q = 5$. However, there is some uncertainty in our extrapolation of $q$. In Figure 26, we show the number density of axions as a function of $q$ before and after the nonlinear regime (at $t = t_\ell$) for $\xi_\star \log_\star = 10^3$, $x_0 = 10$ and $\alpha = 8$. It is clear that provided $q \gg 1$, its actual value introduces only a very minor uncertainty on the final axion number density.

## E.4  Systematics

Systematic uncertainties and numerical artifacts in the axion only simulations can arise from various sources. Here we describe the most important of these, and the choices of simulation parameters that were used to obtain our final results.

First, we note that the number of Hubble patches in the box and the axion mass in lattice units are

$$HL = H_\star L_\star \left( \frac{H}{H_\star} \right)^{1/2} \propto t^{-1/2} , \quad m_a \Delta = \left( \frac{H_\star L_\star}{N} \right) \left( \frac{H_\star}{H} \right)^{\alpha/4 + 1/2} \propto t^{\alpha/4 + 1/2} . \quad (41)$$

The former decreases with time, while the latter increases fast. Therefore the following sources of systematic uncertainty need to be considered.

- The continuum limit corresponds to $m_a \Delta \to 0$, and in particular if $m_a \Delta \gtrsim 1$ lattice effects are introduced. All our simulations are stopped when $m_a \Delta = 1$ so that the discretization effects are negligible for practically the whole simulation time.

- The infinite volume limit corresponds to $HL \to \infty$. In Figure 27 (left) we show that the initial number density is free from finite volume effects provided $H_\star L_\star \geq 1$, which matches expectations based on the form of $F$. Although during the subsequent evolution $HL < H_\star L_\star$, since no strings are present in the simulation, no new IR axion modes are produced and those modes that are present initially still fit in the simulation volume at later times. Therefore, volume effects do not affect the simulation at $H < H_\star$. Instead, the systematic errors introduced by a finite $H_\star L_\star$ even decrease over the course of a simulation. This is because the nonrelativistic regime tends to wash out the dependence on the IR shape of the spectrum, and this is the most affected by finite volume effects. From eq. (41) the time range $H_\star / H$ of a simulation (before $m_a \Delta = 1$ is reached) is

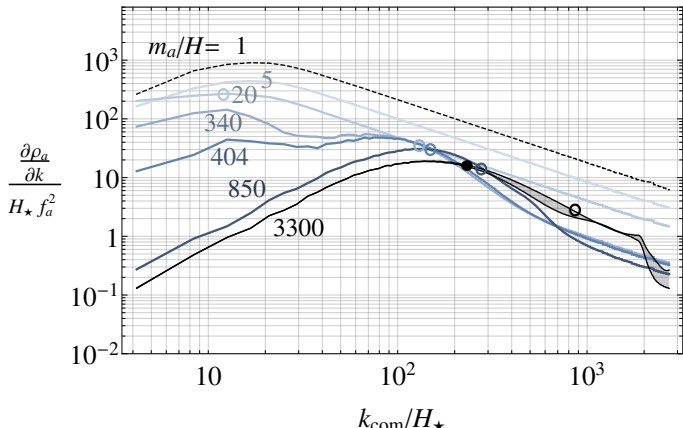

Figure 25: The time evolution of the axion spectrum $\frac{\partial \rho_a}{\partial k}$ for $\xi_\star \log_\star = 1000, \alpha = 8, x_0 = 10$. Different times are represented by $m_a/H$. Results are shown at the initial time $t_\star$ (dashed line), during the relativistic regime (second line from the top), during the transient, and at the final time once the IR modes are evolving nonrelativistically (black line). The analytic fit for time $t_\ell$ when $\rho_{\rm IR} \approx V$ corresponds to $m_a(t_\ell)/H(t_\ell) \approx 680$. Empty dots indicate the momentum equal to $m_a$ at any given time.

maximized by choosing the smallest viable $H_\star L_\star$.[46] Thus, all of our results are obtained from simulations that have $H_\star L_\star \geq 1$.

- The maximum momentum of modes supported in a simulation is only $k_{max} \equiv 2\pi N_x/L$, which is far from the UV-cutoff of the (scale invariant part of the) spectrum, i.e. $k \sim \sqrt{H_\star f_a}$ for the physical parameters. Therefore, given the scale invariance of the convoluted spectrum most of the kinetic energy density of the axion field will not be contained in the grid. However, given the discussion in Section 3, the number density $n_a^{\rm str}$ is dominated by momenta of order $m_a(t_\ell)$ at $t = t_\ell$. Thus, if the UV-cutoff $\Lambda_{UV}$ of the grid satisfies $\Lambda_{UV}(H_\ell/H_\star)^{1/2} \gg m_a(t_\ell)$ the final number density is expected to be independent of $\Lambda_{UV}$.

  We tested the dependence of $n_a^{\rm str}$ on the UV-cutoff by generating initial conditions from the simplified spectrum, but setting $\left.\frac{\partial \rho_a}{\partial k}\right|_\star = 0$ for $k > \Lambda_{UV}$ for different $\Lambda_{UV}$. In Figure 27 (left) we plot the axion number density as a function of time during its mass turn on for different $\Lambda_{UV}$, relative to that of the largest value tested (the simulations are all identical apart from the value of $\Lambda_{UV}$, e.g. they are on the same sized grid). We see that for $\alpha = 8, x_0 = 10$ and $\xi_\star \log_\star = 10^3$ the dependence of $Q$ on $\Lambda_{UV}$ is negligible if $\Lambda_{UV} > 10^3 H_\star$,

  We do however note that as $\xi_\star \log_\star$ increases so does $m_a(t_\ell)$ and therefore a larger UV-cutoff is required. The size of our grids is such that $\Lambda_{UV} > 10^3 H_\star$ in all our simulations, which is sufficient to obtain accurate results.

---

[46]A large time range is needed so that the axion field reaches the nonrelativistic regime, and the asymptotic value of $n_a^{\rm str}$ can be calculated.

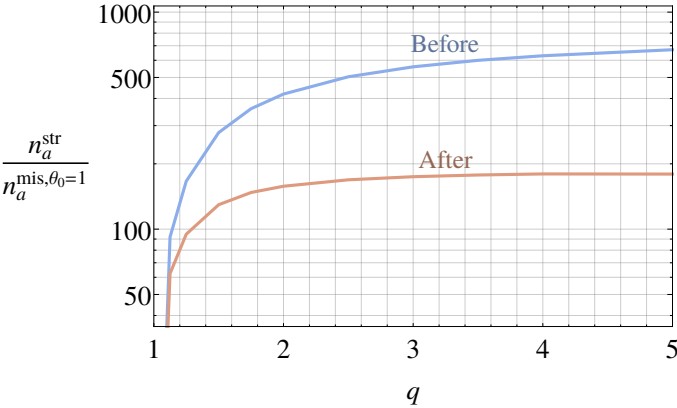

Figure 26: The axion number density from strings relative to that from misalignment, as a function of the power law $q$ of the axion emission spectrum during scaling. We have fixed $\xi_\star \log_\star = 10^3$, $x_0 = 10$ and $\alpha = 8$. The results are shown before and after the axion mass becomes cosmologically relevant.

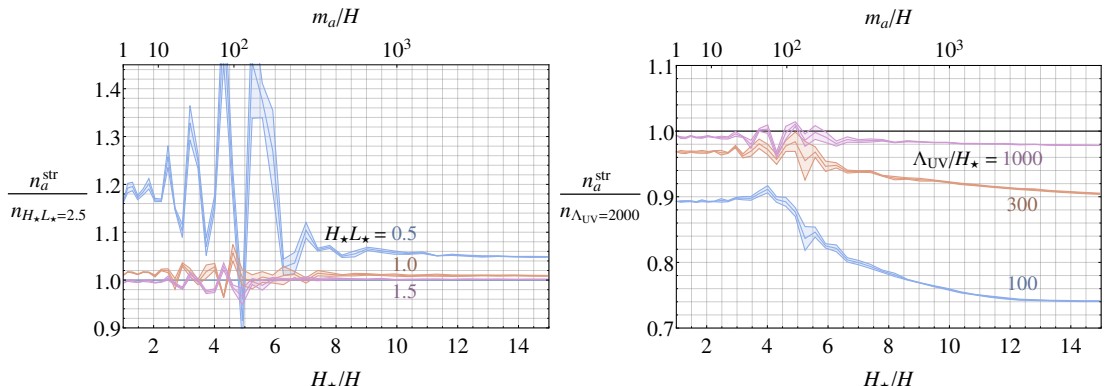

Figure 27: Left: The axion number density as a function of time for different choices of $H_\star L_\star$, normalized to the values closest to the infinite volume limit $H_\star L_\star = 2.5$. Right: the same observable for different choices the UV-cutoff of the spectrum $\Lambda_{UV}/H_\star$, normalized to results with the largest cutoff tested $\Lambda_{UV}/H_\star = 2000$. The simulations are performed for the simplified initial condition spectrum with $x_0 = 10$ and $\xi_\star \log_\star = 10^3$, and an axion mass power law $\alpha = 8$.

## F  Massive Axions on a String Background

In Section 3 we studied the evolution of the axion field as the axion mass becomes cosmologically relevant considering only the axions emitted at earlier times, i.e. for $H > H_\star$. In reality, until the string network is completely destroyed the axion field is made of different components: the axion radiation produced up to $H_\star$ by the scaling regime, axion radiation emitted at later times as the string network is destroyed, and strings themselves. Crucially, in the analysis of Section 3 we implicitly assumed that the presence of strings (which actually store an order one fraction of the energy density in their spatial gradients) does not influence the evolution of the axion radiation, or at least it does not weaken it. In this Appendix we show that the number density of axions from the scaling regime is indeed not affected, at least from the axion string network that can be currently simulated.

## F.1  The Decoupling Limit

For a simulation to include strings, both the axion and the radial mode must be present as dynamical degrees of freedom, and therefore evolved e.g. following the Lagrangian in eq. (18) of Appendix D.[47] An important issue when studying this system is whether the radial mode is sufficiently heavy that it has reached the physical decoupling limit.[48] Indeed, the equations of motion of the complex scalar field, eq. (19), tend to those of the axion, eq. (20), only when the two limits

- $m_a/m_r \to 0$
- $\frac{\partial_\mu(a/f_a)}{m_r} \to 0$ ,

are both satisfied.[49]

The first limit requires that the axion is much lighter than the radial mode, and the second that the typical axion momenta are much smaller than $m_r$. Although this is the case in the physical regime, for which $\log_\star \equiv \log(m_r^\star/m_a^\star) \approx 60 \div 70$ and the spectrum is IR dominated, these limits are unreachable in numerical simulations. In particular, we know that for $\log \lesssim 6 \div 8$ the spectrum is still dominated by axions with momentum of order $m_r$. Moreover, although $m_a/m_r \ll 1$ is satisfied early in simulations when $H_\star/H \ll 1$, it can be violated at the final simulation times when $H_\star/H \gg 1$. This is because, from eq. (20), the axion mass grows fast for nonzero $\alpha$, especially at the physical value $\alpha \approx 8$. Consequently $m_a$ soon becomes close to $m_r$ (which cannot take physically relevant value $\gg H_*$ due to the finite lattice spacing).

To demonstrate the importance of the decoupling limit, we study the simple homogeneous solution of eq. (20) with $a(t_0) = \theta_0 f_a$ and $\dot{a}(t_0) = 0$. This can be compared with the solution of eq. (19) for the full complex scalar field, with $r(t_0) = \dot{r}(t_0) = 0$ and the same initial conditions for $a(t_0)$. To enable comparison with results from simulations that we show in Section F.2, we solve eq. (19) in the fat string case (i.e. with $m_r$ decreasing with time), and we fix the axion mass by choosing $\alpha = 6$ and $\log_\star = 7$.

In Figure 28 we plot the time evolution of the comoving axion number densities for $\theta_0 = 1$ and $t_0 \ll t_\star$ for the theories with and without the radial mode. The comoving number density is given by $n_a/(H_\star f_a^2(H/H_\star)^{3/2})$, where $n_a$ is defined as $\rho_a/m_a$ and $\rho_a$ is the Hamiltonian density in eq. (4). In the same plot we also show the evolution of the radial mode $r/(f_a/\sqrt{2})$. The non-decoupling of the radial mode modifies the oscillations of $a(t)$ generating an unphysical non-conservation of the comoving axion number density, which is already at the level of 20% for $m_a/m_r = 1/3$. At the same time, the radial mode is increasingly displaced from its minimum, acquiring some of the energy that would otherwise be in the axion field. In other words, light enough degrees of freedom coupled to the axion get excited. This simple analysis shows that any prediction from a simulation where the decoupling limit is not reached will be strongly model dependent, and will not reproduce results in the physically important regime.

---

[47]Other UV completions of the axion are also of course possible.

[48]This limit is qualitatively different from the condition $m_a^2/m_r^2 < 1/39$ mentioned in [22], and applies also in the absence of domain walls.

[49]This can be shown by multiplying both sides of eq. (19) by $e^{-i\frac{a}{f_a}}$ and writing $\phi = \frac{r+f_a}{\sqrt{2}}e^{i\frac{a}{f_a}}$. Working in flat spacetime for simplicity, the imaginary and real parts of eq. (19) then become

$$\begin{cases} (1+\sigma)\partial_\mu\partial^\mu\theta + 2\partial_\mu\sigma\partial^\mu\theta + m_a^2\sin\theta = 0 \\ \frac{1}{2m_r^2}\partial_\mu\partial^\mu\sigma - (1+\sigma)\frac{(\partial_\mu\theta)^2}{m_r^2} + \frac{(1+\sigma)}{2}((1+\sigma)^2 - 1) - \frac{m_a^2}{m_r^2}\cos\theta = 0 \end{cases} , \quad \sigma \equiv \frac{r}{f_a/\sqrt{2}}, \quad \theta \equiv \frac{a}{f_a}. \quad (42)$$

Eq. (42) with the radial mode on the VEV, $r = 0$, reduces to the axion equation (20) only in the limit $m_a/m_r \to 0$ and $\partial_\mu\theta/m_r \to 0$, so that the second equation is trivially satisfied and the first reduces to eq. (20). If this limit is not reached, the second and fourth terms in the second equation act as a source for the radial mode, which is then generated even starting from $r = 0$.

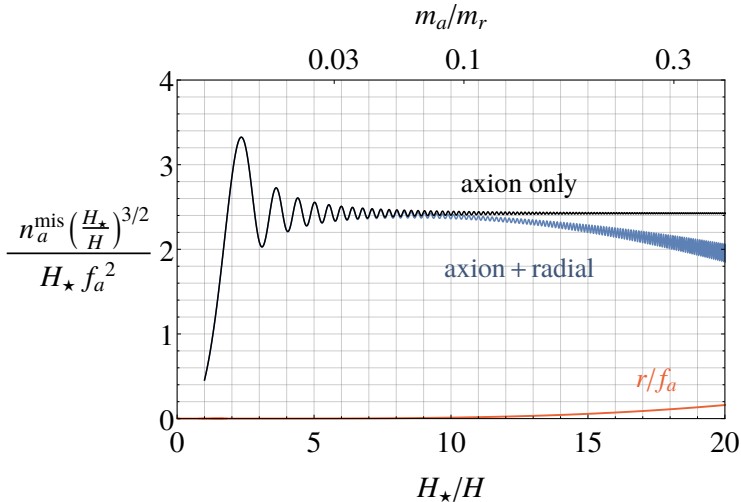

Figure 28: A comparison between the comoving axion number density from the homogeneous misalignment with $a(t_0) = f_a$ using the axion only equations (black) and the full complex scalar field equations with $\log(m_r^\star/m_a^\star) = 7$ (blue). The amplitude of the radial mode $r/(f_a/\sqrt{2})$ is also shown. Already at $m_a/m_r = 1/3$, the radial mode is unphysically displaced from the minimum and the axion number density is not conserved by $\approx 20\%$.

## F.2 The Effect of Strings

Next we test whether the presence of strings affects the dynamics of the pre-existing radiation, and therefore the number density of axions resulting from the scaling regime. To do so we simulate the fat string system starting from its evolution during the scaling regime, through the time when the axion mass turns on. At $H = H_\star$ we modify the complex scalar field by injecting additional axion radiation, with the spectrum that would be produced at $\log_\star = 60 \div 70$ from the scaling regime with $q > 1$.

The extra radiation must be introduced to account for emission by the scaling regime at large values of the log and to enable a fair comparison with Section 3. In contrast, the radiation component emitted by the string network prior to $\log_\star \approx 7$, which is directly accessible in simulations, is still UV-dominated. This will therefore not capture the dynamics that we are interested in, and will probably make a negligible contribution to the axion number density compared to misalignment. Of course, with this setup we do not aim to compute the complete axion relic abundance from strings and domain walls. Instead we simply want to confirm that our analysis in Section 3 is not significantly altered by the actual presence of strings and domain walls, and confirm that our analysis does indeed give a lower bound on the relic abundance.

To do so, we solved eq. (19) as in Appendix D starting from $H = m_r$. Then when $t = t_\star$, in the middle of the evolution, we substituted

$$\phi(t_\star, \vec{x}) \to \phi(t_\star, \vec{x}) e^{i a_w(t_\star, \vec{x})/f_a} \quad \text{and} \quad \dot{\phi}(t_\star, \vec{x}) \to \frac{d}{dt}\left(\phi(t, \vec{x}) e^{i a_w(t, \vec{x})/f_a}\right)\Bigg|_{t=t_\star} . \quad (43)$$

The field $a_w(t_\star)$ is the additional radiation and is extracted from the kinetic energy spectrum $\frac{\partial \rho_a}{\partial k}\Big|_\star$ of the scaling regime at $\log_\star = 60 \div 70$ (as in Appendix E.2).

There are a number of potential sources of systematic errors to be taken into account in such simulations, on top of those already present in the analysis of the scaling regime for vanishing axion mass.

- The ratio $m_r/m_a$ should be large enough at the final time not to introduce the non-decoupling effects discussed in the previous Section. In particular, even if we will use $m_r\Delta = 1$ as in the string network simulations, $m_a\Delta = 1$ at the final time will not be sufficient. Instead we stop the simulations at $m_r/m_a = 10$ (Figure 28 suggests that this is sufficiently large to avoid unphysical energy transfer to the radial mode, although it is not definitive since that plot is for a homogeneous field).

- $H_\star L_\star$ should be large enough so that finite volume effects do not cause the string network to shrink. Instead we want this to occur due to the the axion mass. The resulting constraint on $H_\star L_\star$ is stronger than that discussed in Appendix E.2). For $\alpha = 6$, we checked that the scaling parameter has dropped by 50% due to the mass at $H_\star/H = 4$, so choosing $H_\star L_\star \gtrsim 4$ is sufficiently safe for our purposes.

- The UV cut-off of the injected spectrum (which is scale invariant) should be smaller than $m_r/2$ to satisfy the second requirement for the decoupling limit. We cutoff the injected spectrum for momenta bigger than $k_{\mathrm{UV}} = 50 H_\star$ (as is described in Appendix E.4). For $\log_\star = 7$ this is sufficiently small compared to $m_r$ not to introduce major effects.[50]

Accommodating the competing requirements of these sources of systematic errors is challenging. Simulations cannot last as long as the axion-only simulations of Section 3, and we cannot reach the regime where the comoving number density is precisely conserved (after the relativistic period and the nonlinear transient). This is the case even when a spectrum with relatively small $\xi_\star \log_\star = 200$ is injected, for which the nonrelativistic regime is reached relatively early.

Despite these difficulties, results from simulations are still sufficient to show that the presence of strings does not affect the existing radiation, enough for our present work. In Figure 29 we plot (in blue) the evolution of the axion number density when the axion spectrum corresponding to $\xi_\star \log_\star = 200$ is injected into a simulation with strings at the simulation time $\log = \log_\star = 7$.[51] As in Figure 28 we take $\alpha = 6$. More precisely, we calculate the axion number density from the kinetic component of its energy, i.e. in the first term of eq. (39), multiplied by a factor of two.[52] This gives the correct result at the early and late times, and during the transient the axion number density is not well defined anyway.

For comparison, in Figure 29 we also plot in pink the number density calculated in an axion-only simulation of Section 3 with the same initial spectrum.[53] Additionally, we plot in orange the number density when the axion spectrum is injected into a simulation of the complex scalar field, but with no strings or preexisting radiation present. Finally, we show the number density arising from directly simulated strings when the axion spectrum is not injected.

The effect of the non-decoupling of the radial mode can be seen in these results. The final number density from simulations involving the radial mode is somewhat smaller than that from an axion only simulation, even when there are no strings or radiation present. This happens because we do not have the numerical power to simulate a very large hierarchy between $m_a$ and $m_r$ at such times. As expected the number density injected is much larger than that emitted

---

[50]If substantially larger values of $k_{\mathrm{UV}}$ are used the field evolution develops numerical instabilities.

[51]The injected spectrum has a UV cutoff at $k/H = 50$ to maintain the required hierarchy $k \ll m_r$. We choose $\xi_* \log_* = 200$, somewhat smaller than our central value for the QCD axion, since simulations with larger values require smaller lattice spacing, increasing the computational resources required.

[52]We do this because the presence of strings and oscillons makes it challenging to evaluate the axion field numerically without complications from discontinuities, due to its periodic nature. However, the time derivative of the axion can easily be computed.

[53]Unlike Appendix E in these we calculate the number density from twice the kinetic component to enable comparison with simulations of the complex scalar field.

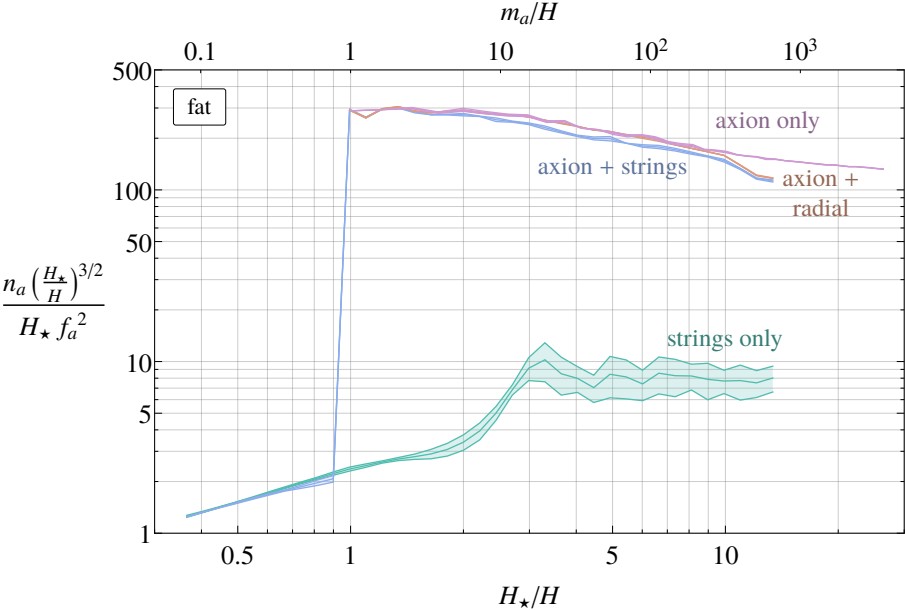

Figure 29: The evolution of the axion number density (extracted as twice the kinetic component of the number density) through the axion mass turn on, for $\alpha = 6$. We compare simulations in which only the axion is dynamical (pink), starting with the initial field configuration predicted from the scaling regime with $\xi_* \log_* = 200$, with results when the same axion spectrum is injected into a simulation of the axion and radial mode that has a string background evolved with a potential such that $\log_\star = 7$ (blue) (via eq. (19)). In the latter, the string network is in the scaling regime prior to the axion mass turn on, and is subsequently destroyed. We also plot the evolution of the same axion spectrum injected into a simulation of the axion and radial mode with no strings present (orange), and the result from the scaling regime without the addition of the extra axion field (light blue). The agreement between the results when the axion spectrum is evolved in complex scalar field simulations with and without strings shows that the presence of strings with $\log_\star = 7$ does not have a significant effect on the dynamics of the preexisting radiation.

by strings, during scaling and also as they are destroyed by the axion mass, at the values of $\log(m_r/H)$ that can be simulated

Most importantly, the final axion number density when the spectrum is injected into a simulation with strings is very close to that when it is injected into a simulation of the complex scalar with no strings. This indicates that the presence of strings, and the dynamics of their annihilation at the end of the scaling regime, does not significantly alter the evolution of the pre-existing axion radiation. Since the number density from the axions in the scaling regime is not depleted by strings with $\log_\star = 7$, there is no reason to expect that this will not also hold for larger $\log_\star$. Indeed is seems implausible that strings could absorb all of the energy in axions previously released by the scaling regime, and then emit this back as high momentum axions, which would be required for the strings to decrease the final axion number density.

# G  Comparisons

Our final conclusions in Section 4 differ from those obtained by other authors in the literature for various reasons. Here we comment only on those that are relatively close to us in their

basic assumptions or techniques used.

- Based on the expected similarity between axion strings at $\log_\star \gg 1$ and Nambu–Goto strings, the authors in [33–36] assumed values for $\xi_\star$ and $q$ numerically compatible with those inferred by our study, and therefore got an enhanced contribution to the axion abundance from topological defects. Their analysis however did not take into account the effects from nonlinearities induced by the large axion field values. These, as we have shown, crucially affect the field evolution during the QCD transition and substantially change the final axion abundance.

- In [25] the authors performed a simulation of the entire axion string/domain-wall system's evolution from the scaling regime until the linear regime after the QCD transition and beyond. The result for the abundance is substantially smaller than the bound in eq. (6), however it is not in contradiction with it. Indeed the range of $\log(m_r/H)$ that can be directly simulated does not allow large values of $\xi_\star \log_\star$ to be reached (and instead $\xi_\star \log_\star$ remains a couple of orders of magnitude below the physical one), nor does it allow the spectrum to be seen turning IR dominated (i.e. with $q > 1$). Consequently, the number of axions produced by strings will be small, thus the smaller abundance measured in such a simulation. In studying the effect of strings and walls in the evolution of the axion field during the nonlinear regime in Appendix F we also performed simulations analogous to those of ref. [25] obtaining compatible results. However from Fig. 29 in Appendix F.2 it is clear that ignoring a proper extrapolation of the parameters could easily lead to the dominant contribution to the abundance being missed and the total contribution being underestimated by more than one order of magnitude.

- As in the previous case, the authors of ref. [93] perform simulations of the entire evolution from the scaling regime to the linear regime including the decay of strings and walls. However, by changing the physics at the string core scale $m_r$ they manage to produce an effective string tension which is numerically equivalent to $\log_\star \simeq 70$. In doing so they also observe an enhancement of $\xi_\star$ which grows by a factor of a few. At $t_\star$ the system has an effective energy density prefactor $\xi_\star \log_\star$ of the same order of magnitude of our extrapolated one. Despite the large energy density, the final axion abundance found is small as if the string contribution was negligible. The result is not in obvious contradiction with our findings and can be understood as follows. Within the range of $m_r/H$ that can be simulated we observed that most of the energy of the string network is still dumped into UV modes, which have not yet decoupled and instead influence the IR string dynamics. Despite the higher string tension, the evolution of the string network in ref. [93] is probably still dominated by UV modes (which in such a setup are unphysical) producing a spectrum with $q < 1$ explaining the suppressed axion abundance. Unfortunately in order to check this interpretation of the disagreement a dedicated study of the axion spectrum in this setup is required, which is currently missing.

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
