# Peer review of "More Axions from Strings"

_SciPost Physics, doi:SciPost Phys. 10, 050 (2021)_

## Round 1 · Referee Report · Anonymous (Referee 1) · 2020-11-23

Report

The paper addresses the contribution of axion strings to the axion DM relic density. This is one of the most important open questions in axion physics, since a theoretical determination of the axion DM mass (in the post-inflationary PQ breaking scenario) would suggest where to first focus experimental searches in an otherwise vast parameter space. Although many researchers have previously dealt with this problem, a reliable answer is prevented by the need to carry on numerical simulations over a huge range of scales.

Based also on their previous work (Ref. [7]) the authors have performed extensive numerical simulations and provided analytical insight for extrapolating the results to the physical point. Given the impossibility of reliably assessing the full contribution of topological defects (strings and domain walls), they focus solely on the contributions of axions radiated from strings in the so-called scaling regime, before the axion potential becomes relevant. This provides a lower-bound on the axion DM mass that turns out to be surprisingly stronger (by more than one order of magnitude) than the standard contribution from the misalignment mechanism. This is a somewhat unexpected result which has potentially game-changing consequences for axion searches.

The ingredients leading to such conclusions are two-fold: the evidence for logarithmic violations of the scaling regime and for a non-trivial running of the spectral index, that is more compatible with an IR dominated spectrum. These results also imply that by the time the axion mass turns on the amplitude of axion radiation produced by strings is large, so that non-linear effects need to be taken into account for the final estimate of the number density of axion from strings. These conclusions are convincingly supported both via numerical simulations and analytical arguments.

The paper is also well structured: the main results are summarised in the body, while further details (especially regarding the simulations) are deferred to 7 Appendices. The last Appendix provides a comparison with other works in the literature. This is very helpful, given the the fact that results are often discordant and possible origins for such discrepancies are analysed and highlighted.

The paper is outstanding both for the rigour of the methodology and the impact of the results.

Requested changes

I have some comments/questions. The authors can decide themselves if they want to further address them:

1-It is not obvious how to properly rescale the results for different values of the domain wall number (N). To this end, it would be helpful to have a plot displaying the lower bound on the axion DM mass as a function of N. This could bear some interest in the case of large N models addressing the PQ quality issue.

2-The contribution from topological defects to the axion number density acquires a non-trivial dependency from UV physics (differently from the misalignment contribution). While the fact that the axion spectrum turns out to be IR dominated makes the final result solid against the details of UV physics, I wonder whether there are situations in which this is not the case. For instance, what would happen if the radial mode is much ligher than $f_a$ (as in the case classically scale invariant setups)?

3-How to make further progress in the field? Is there a conceivable strategy to reliably include the missing contribution from the string-domain wall system?

---

## Round 1 · Referee Report · Anonymous (Referee 2) · 2021-2-4

Report

The authors study the cosmology of the QCD axion for the case where PQ breaking occurs after inflation, with a particular focus on axion production from topological defects. A direct computation of the nonlinear evolution of the string-domain wall system and the resulting axion abundance is extremely difficult due to the vastly different scales involved. This work therefore concentrates on the (still technically very demanding) regime where an approximate scaling solution holds to derive a
conservative lower bound on the resulting axion abundance.
This directly translates to a lower bound on the axion mass which is significantly above the value of the naive misalignment prediction.

This is a very important result that directly impacts the search regions and expectations with regards to axion searches.
I therefore completely agree with the first referee that this is an outstanding piece of work that clearly deserves publication.

---

## Editorial Decision

published